# Plekhg5 controls the unconventional secretion of Sod1 by presynaptic secretory autophagy

Amy-Jayne Hutchings[1,16], Bita Hambrecht[1,16], Alexander Veh [1,16], Neha Jadhav Giridhar [1], Abdolhossein Zare [1], Christina Angerer[1], Thorben Ohnesorge[1], Maren Schenke [2,3], Bhuvaneish T. Selvaraj[4,5,6], Siddharthan Chandran [4,5,6], Jared Sterneckert[7,8], Susanne Petri[9], Bettina Seeger [2], Michael Briese [1], Christian Stigloher [10], Thorsten Bischler [11], Andreas Hermann [12,13,14], Markus Damme [15], Michael Sendtner [1] & Patrick Lüningschrör [1] ✉

Increasing evidence suggests an essential function for autophagy in unconventional protein secretion (UPS). However, despite its relevance for the secretion of aggregate-prone proteins, the mechanisms of secretory autophagy in neurons have remained elusive. Here we show that the lower motoneuron disease-associated guanine exchange factor Plekhg5 drives the UPS of Sod1. Mechanistically, Sod1 is sequestered into autophagosomal carriers, which subsequently fuse with secretory lysosomal-related organelles (LROs). Exocytosis of LROs to release Sod1 into the extracellular milieu requires the activation of the small GTPase Rab26 by Plekhg5. Deletion of *Plekhg5* in mice leads to the accumulation of Sod1 in LROs at swollen presynaptic sites. A reduced secretion of toxic ALS-linked SOD1$^{G93A}$ following deletion of *Plekhg5* in SOD1$^{G93A}$ mice accelerated disease onset while prolonging survival due to an attenuated microglia activation. Using human iPSC-derived motoneurons we show that reduced levels of PLEKHG5 cause an impaired secretion of ALS-linked SOD1. Our findings highlight an unexpected pathophysiological mechanism that converges two motoneuron disease-associated proteins into a common pathway.

Autophagy is an essential cellular degradation pathway. In neurons, autophagy contributes to the maintenance of global proteostasis[1,2] but is also tightly linked to activity-related pre- and postsynaptic processes[3–6]. Besides its well-characterized role in feeding lysosomes with cargo material for degradation, there is an emerging role of autophagy in unconventional protein secretion (UPS)[7]. Cytosolic proteins that do not contain an N-terminal signal sequence for conventional secretion via the ER and Golgi are released from cells by UPS[8]. During secretory autophagy, autophagosomes serve as vesicle carriers to deliver these proteins to the extracellular milieu[7]. However, before fusing with the plasma membrane, the autophagosomes may transit through the endo/lysosomal pathway and fuse with multi-vesicular bodies, endosomes, or lysosomes, possibly generating a vast diversity of secretory vesicle carriers[9]. Recent work has linked the UPS to the secretion of aggregate-prone proteins, such as amyloid beta and hyperphosphorylated tau in Alzheimer's disease, mutant huntingtin in Huntington's disease, and superoxide dismutase 1 (SOD1) and TDP43 in Amyotrophic Lateral Sclerosis (ALS)[10–15]. However, the molecular mechanisms of UPS and secretory autophagy are poorly understood.

Pleckstrin homology domain-containing family member 5 (PLEKHG5) is a guanine exchange factor (GEF) predominantly expressed in the nervous system[16]. Plekhg5 regulates the autophagy of synaptic vesicles by activating Rab26, a small GTPase enriched on synaptic vesicles[17,18]. Mutations in the human *PLEKHG5* gene have been linked to several forms of motoneuron disease (MND)[19–26]. Initially, a single homozygous missense mutation had been identified in an individual extended consanguineous African family diagnosed with a unique form of autosomal recessive lower MND with childhood-onset (distal spinal muscular atrophy, DSMA4)[24]. More recently, additional *PLEKHG5* mutations were frequently identified in patients suffering from autosomal recessive intermediate Charcot-Marie-Tooth disease (CMT), DSMA, and distal hereditary motor neuropathy (dHMN)[19,25,26]. Taken together, *PLEKHG5* has emerged as a critical factor in preserving the integrity and function of motoneurons (MNs) and the peripheral nervous system. However, besides its role in synaptic vesicle autophagy, not much is known about the function of Plekhg5.

Here, we present an unexpected link between Plekhg5 and Sod1, one of the most frequently mutated proteins in familial ALS (fALS)[27]. We show that Sod1 accumulates within axon terminals in the spinal cord and MN terminals in the skeletal muscle of Plekhg5-deficient mice. Plekhg5 depletion in cultured MNs and NSC-34 cells resulted in a reduced UPS of Sod1. Mechanistically, our data suggest that Sod1 is translocated into an autophagosomal vesicle intermediate, which fuses with a lysosomal-related organelle (LRO) before fusion with the plasma membrane. This secretory pathway depends on the activation of the small GTPase Rab26 by Plekhg5. To study the relevance of this mechanism for the pathophysiology of MND, we crossed Plekhg5-deficient mice with SOD1[G93A] ALS mice. Depletion of Plekhg5 from SOD1[G93A] mice resulted in prolonged survival but an accelerated disease onset. Using RNAseq and histology, we identified an attenuated microglial activation in Plekhg5-depleted SOD1[G93A] due to the reduced binding of extracellular mutant SOD1[G93A] to microglia. Moreover, we used ALS-patient-derived induced pluripotent stem cells (iPSCs) to study the secretion of mutant ALS-linked SOD1 expressed at physiological levels. We found that ALS-linked SOD1 MNs displayed a reduced secretion of mutant SOD1. Notably, the reduced secretion coincided with reduced expression levels of PLEKHG5, which could be restored by overexpression of PLEKHG5.

In summary, we highlight an unexpected pathophysiological mechanism involving secretory autophagy. Disruption of this pathway by Plekhg5 depletion results in deleterious vesicle accumulation, including Sod1. The interplay between Plekhg5 and Sod1 connects two different facets of MNDs into one coherent pathway.

## Results

### Plekhg5 depletion results in intracellular presynaptic Sod1 accumulations

Previous work found that Plekhg5 depletion in mice leads to synaptic vesicle accumulations in MN terminals and peripheral nerves of Plekhg5-deficient mice without affecting global proteostasis[17,28]. However, whether synaptic vesicles are the only cargo material affected by Plekhg5-dependent autophagy has not been explored.

We analyzed the vesicle accumulations in Plekhg5-deficient mice in more detail and found, to our surprise, intracellular accumulations of endogenous Sod1 in spinal cord cross-sections of Plekhg5-deficient mice (Fig. 1A, B). Notably, these accumulations appeared without overexpression of human ALS-linked mutant SOD1. We did not detect any accumulations of other aggregation-prone proteins such as Tau, TDP43, or the autophagy-adaptor p62, suggesting that Plekhg5 specifically affected the turnover of Sod1 (Supplementary Fig. 1). Sod1 accumulations were present throughout the different areas of the spinal cord, including the ventral horn (Fig. 1A). We also detected the clustering of Sod1 within axon terminals of MNs at neuromuscular junctions (NMJs) (Fig. 1C). However, Sod1 accumulations were absent from the somata of neurons (Fig. 1D). To confirm the neuronal origin of the Sod1 accumulations within the spinal cord, we crossed Plekhg5-deficient mice with Thy1::YFP mice to label individual neurons. Using this strategy, we detected Sod1 accumulation in proximal and distal parts of YFP-labeled axons, confirming the intracellular localization of these accumulations in axons (Fig. 1E, F). At high resolution, the Sod1 accumulations appeared as clusters of individual Sod1[+] structures (Fig. 1G). To assess whether the Sod1 accumulations represent an insoluble aggregated species of Sod1, we biochemically separated the insoluble proteins from the Triton-X 100 soluble fraction and analyzed the Sod1 levels in both fractions by Western blot (Fig. 1H). We detected significantly elevated levels of Sod1 in Triton-X 100 soluble fractions but not in the insoluble fractions (Fig. 1I). Collectively, these data suggest that Plekhg5 depletion causes the accumulation of Sod1 in the presynaptic compartment.

### Plekhg5-mediates the unconventional secretion of Sod1 via Rab26

Next, we investigated how Plekhg5 depletion affects the turnover of Sod1. Previous studies showed that Sod1 is sequestered by the autophagosomal/lysosomal pathways but also by the proteasome[29–32]. However, Sod1 is a cleavage-resistant protein, which appeared stable in lysosomes compared to carbonic anhydrase III, another cytosolic protein[32]. Furthermore, Sod1 can be secreted from cells[14,33]. Therefore, we analyzed the levels of Sod1 in cell lysates and medium of both cultured primary MNs and NSC34 cells upon sh-RNA-mediated depletion of Plekhg5 (Fig. 2A–E). In order to analyze the secretion of Sod1, we utilized a previously established protocol to examine the ratio between Sod1 in the medium and the corresponding cell lysate by Western blot[14]. As previously described, we detected small fractions of total Sod1 in the medium[14]. Upon knockdown of Plekhg5, we observed significantly reduced levels of secreted Sod1 and correspondingly elevated Sod1 levels in cell lysates in both cell types. (Fig. 2B–E). Our immunohistochemical stainings suggest that Sod1 predominantly accumulates in the presynaptic compartment in Plekhg5-deficient mice. To examine whether Plekhg5 impairs the Sod1 secretion in axons in vitro, we cultured primary MNs in compartmentalized chambers and analyzed the Sod1 levels in the media and lysates of the somatodendritic and axonal compartment (Fig. 2F–H). In the axonal compartment, we detected a marked increase in the ratio between Sod1 in the medium and lysate, indicating a more efficient secretion of Sod1 in axons (Fig. 2F, H). Upon depletion of Plekhg5, we observed a significant reduction of Sod1 in the media of the axonal compartment and an increase of Sod1 in lysates on the somatodendritic side (Fig. 2G, H). From these data, we conclude that the Sod1 secretion is more efficient in axons and that Plekhg5 depletion blocks the secretion of Sod1 in axons.

We previously showed that Plekhg5 mediates the autophagy of synaptic vesicles by Rab26[17]. Besides directing synaptic vesicles to autophagosomal structures, Rab26 has been linked to different secretory processes in several cell types[34–37]. Therefore, we examined the involvement of Rab26 in the UPS of Sod1. First, we knocked down Rab26 using two different sh-RNAs in primary MNs and analyzed the levels of Sod1 in lysates and the corresponding media (Fig. 2I–K). Both knockdowns resulted in reduced levels of Sod1 in the media (Fig. 2J, K). Interestingly, the effect size correlated with the efficiency of the knockdown (Fig. 2I, K). To functionally link Rab26 and Plekhg5, we expressed wildtype and a constitutively active, GTP-locked version of Rab26 (RAB26-Q123L, referred to as Rab26-QL) in Plekhg5-depleted MNs, and analyzed the UPS of Sod1 (Fig. 2L–N). Whereas wildtype Flag-Rab26 did not alter the medium levels of Sod1, expression of Flag-Rab26-QL restored the levels of Sod1 in the medium of Plekhg5-depleted primary MNs (Fig. 2M, N). In summary, these results suggest that Plekhg5 mediates the secretion of Sod1 by activating the small GTPase Rab26.

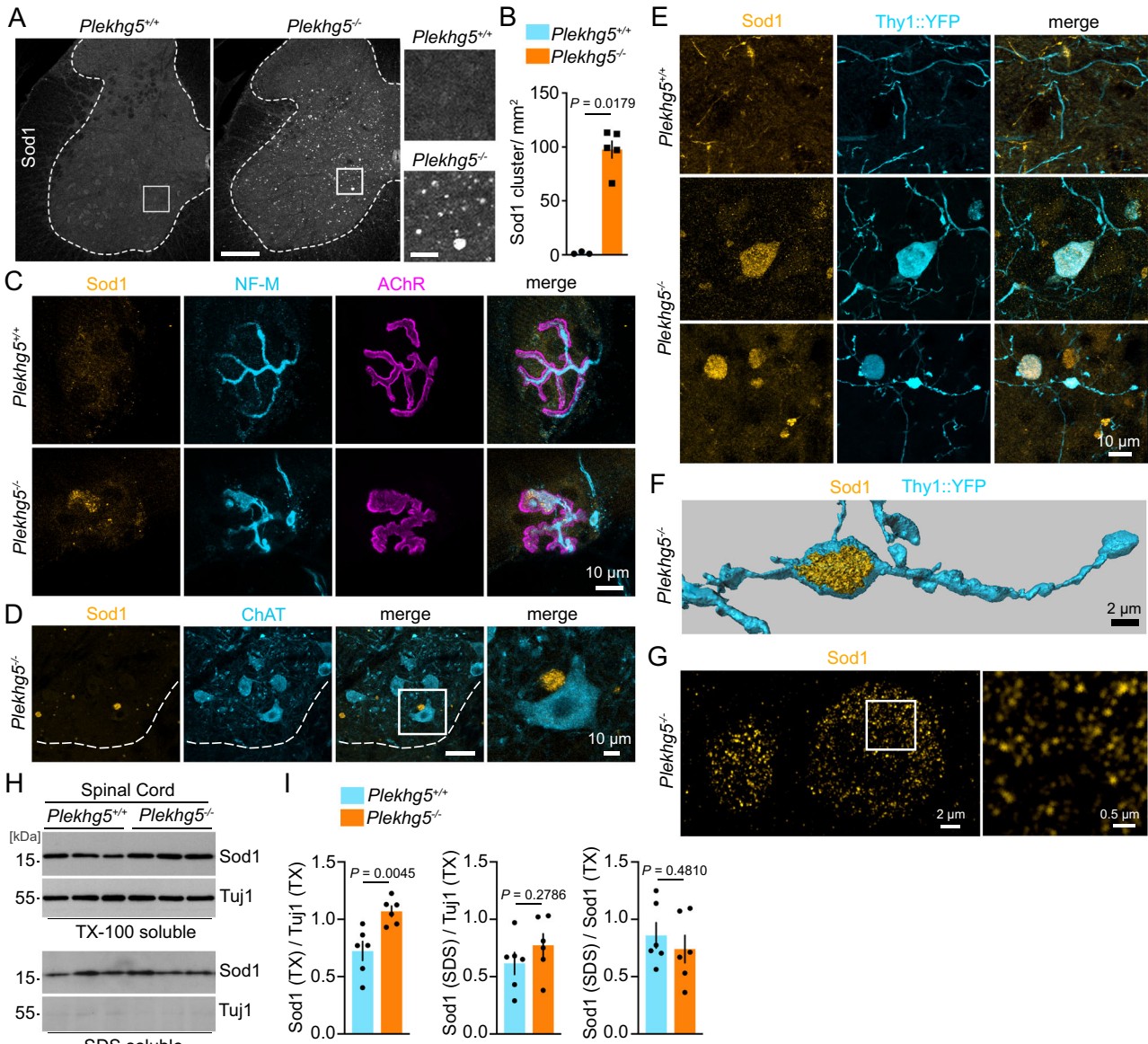

**Fig. 1 | Plekhg5 regulates the secretion of Sod1. A** Representative images of spinal cord cross sections from *Plekhg5⁺/⁺* and *Plekhg5⁻/⁻* mice stained for Sod1. Numerous accumulations of Sod1were present in spinal cord cross sections from *Plekhg5⁻/⁻* mice. Scale bar: 200 μm. Scale bar, inset: 40 μm. **B** Quantification of Sod1 accumulations in spinal cord cross sections. *Plekhg5⁺/⁺* mice, n = 3; *Plekhg5⁻/⁻* mice, n = 5; biological replicates. 7 sections were analyzed per mouse. Mann-Whitney test, one-tailed. **C** NMJ within the gastrocnemius visualized by BTX and NF-M staining. Note the Sod1 accumulation at the axon terminal of *Plekhg5⁻/⁻* mice. The images are representative of at three biological replicates. **D** Spinal cord cross-section of *Plekhg5⁻/⁻* stained for Sod1 and ChAT showing a Sod1 accumulation close to the soma of a MN. The images are representative of two biological replicates. Scale bar: 40 μm. Scale bar, inset: 10 μm. **E** Spinal cord cross-sections of *Plekhg5⁻/⁻ Thy1::YFP* mice stained for Sod1 and GFP reveal the localization of Sod1 accumulations in distal parts of axons. The images are representative of three biological replicates. **F** Imaris 3D reconstruction of the image shown in the lower panel of **E. G** SIM microscopy of Sod1 accumulations in the spinal cord of Plekhg5-deficient mice. At high resolution Sod1 accumulations appeared as clusters of individual Sod1⁺ structures. The images are representative of three biological replicates. **H** Accumulation of Triton-X-100 soluble Sod1 in spinal cords of *Plekhg5⁻/⁻* mice. Spinal cord homogenates were separated into Triton-X-100-soluble and SDS-soluble fractions and analyzed by Western blot. **I** Quantification of the Sod1 levels in the Triton-X-100-soluble and SDS-soluble fractions of *Plekhg5⁻/⁻* and *Plekhg5⁺/⁺* mice. n = 6 biological replicates; t-Test, two-tailed. Data are mean ± SEM. Source data are provided as a source data file.

## The secretion of Sod1 in motoneurons depends on autophagy

Previously, different pathways for the secretion of overexpressed wildtype and mutant ALS-linked Sod1 have been proposed[14,33,38]. It has been suggested that Sod1 is secreted in an exosome-dependent manner[33,38] or via UPS following a pathway similar to Acb1[14]. Due to its stability in lysosomes[31,32], we hypothesized that Sod1 might be released by secretory autophagy to prevent the accumulation in lysosomes and to maintain the proteostasis, as recently suggested[39].

Treatment with the autophagy inhibitor 3-MA caused a significant reduction of basal Sod1 secretion in NSC-34 cells

(Fig. 3A, C). In contrast, inhibition of exosome release using the neutral sphingomyelinase inhibitor GW4869 had no effects on the Sod1 secretion (Fig. 3B, C). In agreement with previous results, we detected a marked increase in the Sod1 secretion upon starvation-induced autophagy using HBSS (Fig. 3D)[14]. To determine whether Sod1 secretion requires fully formed autophagosomes, we blocked different steps during autophagy biogenesis in primary MNs (Fig. 3E–L). Previous work showed that depletion of Atg9 blocks the generation of the isolation membrane[40]. Lentiviral knockdown of Atg9 in cultured MNs resulted in reduced medium levels of

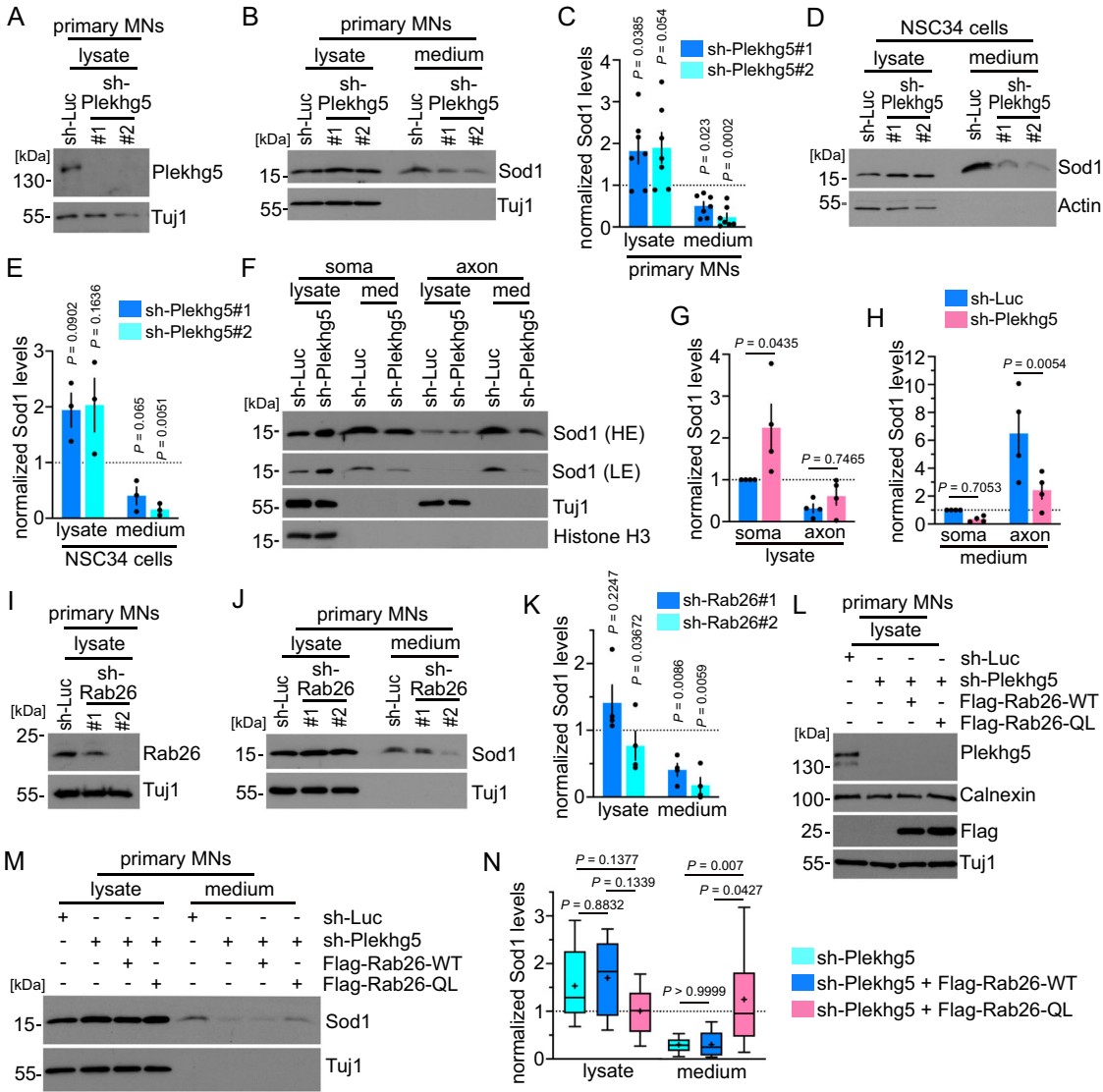

**Fig. 2 | Plekhg5-mediates the unconventional secretion of Sod1 via Rab26.**
**A** Western blots showing the knockdown of Plekhg5 in MNs (sh-RNA-1 = #1; sh-RNA-2 = #2). **B** Reduced Sod1 levels in the medium of Plekhg5-depleted MNs as shown by Western blot. **C** Quantifications of the Sod1 intensities upon knockdown of Plekhg5 in MNs. $n = 7$ biological replicates. One sample t-test, two-tailed. **D** Western blots showing reduced Sod1 levels in the medium of Plekhg5-depleted NSC34 cells. **E** Quantifications of the Sod1 intensities upon knockdown of Plekhg5 NSC34 cells. $n = 3$ biological replicates. One sample t-test, two-tailed. **F** Western blot showing the Sod1 levels in the somatodendritic and axonal compartments upon Plekhg5 knockdown. LE low exposure, HE high exposure. Quantifications of the Sod1 intensities in lysates (**G**) and media (**H**) of the somatodendritic and axonal compartments upon knockdown of Plekhg5 in MNs. $n = 4$ biological replicates. Two-Way ANOVA, Šídák's multiple comparisons test. **I** Western blot showing the knockdown of Rab26 in MNs (sh-RNA-1 = #1; sh-RNA-2 = #2). **J** Western blots showing reduced Sod1 levels in the medium of Rab26-depleted MNs.
**K** Quantifications of the Sod1 intensities upon knockdown of Rab26 in MNs. $n = 4$

biological replicates. One sample t-test, two-tailed. **L** Western blot of MN lysates showing the knockdown of Plekhg5 and simultaneous expression of Flag-Rab26-WT or Flag-Rab26-QL. **M** Expression of Flag-Rab26-QL restores the reduced Sod1 medium levels after Plekhg5 knockdown as shown by Western blot.
**N** Quantifications of the Sod1 intensities upon knockdown of Plekhg5 and simultaneous expression of Flag-Rab26-WT or Flag-Rab26-QL in MNs. sh-Luc, $n = 12$; sh-Plekhg5-E, $n = 12$; sh-Plekhg5-E+Flag-Rab26-WT, $n = 5$, sh-Plekhg5-E+Flag-Rab26-QL, $n = 12$; biological replicates. One-way ANOVA; Tukey's multiple comparisons test. Box bounds are defined by the 25th and 75th percentiles. Extending whiskers represent data points within 1.5 times the interquartile range from lower and upper quartiles. Lines and crosses denote the median and mean. Sod1 secretion was calculated as the ratio between the amount of Sod1 in the medium and in the lysate. Sod1 levels in the lysates were adjusted to Tuj1 (MNs) or Actin (NSC34 cells). The normalized Sod1 intensities were set to 1 in each experiment Data are mean ± SEM. Source data are provided as a Source Data file.

Sod1, without affecting the levels of LC3-II under basal conditions, which is in agreement with previous work (Fig. 3E–H)[41]. We also detected significantly reduced Sod1 medium levels upon Cre-mediated depletion of Atg5 from cultured MNs (Fig. 3I–L). Depletion of Atg5 resulted in a significant reduction of lipidated LC3 (LC3-II) (Fig. 3I, J), which indicates that the Sod1 secretion also requires later steps in the autophagosome biogenesis up to the lipidation of LC3.

To further examine if autophagy directly drives Sod1 secretion, we applied a membrane fractionation procedure to enrich autophagosomal membranes as described previously (Fig. 3M)[42]. First, we performed a differential centrifugation to obtain 3 K, 25 K, and 100k membrane pellet fractions (Fig. 3N). Both Sod1 and LC3-II were mainly enriched in the 25k fraction along with Lamp1 (Fig. 3N). Further separation of the 25k membrane fraction through a sucrose step gradient ultracentrifugation showed that both Sod1 and LC3-II

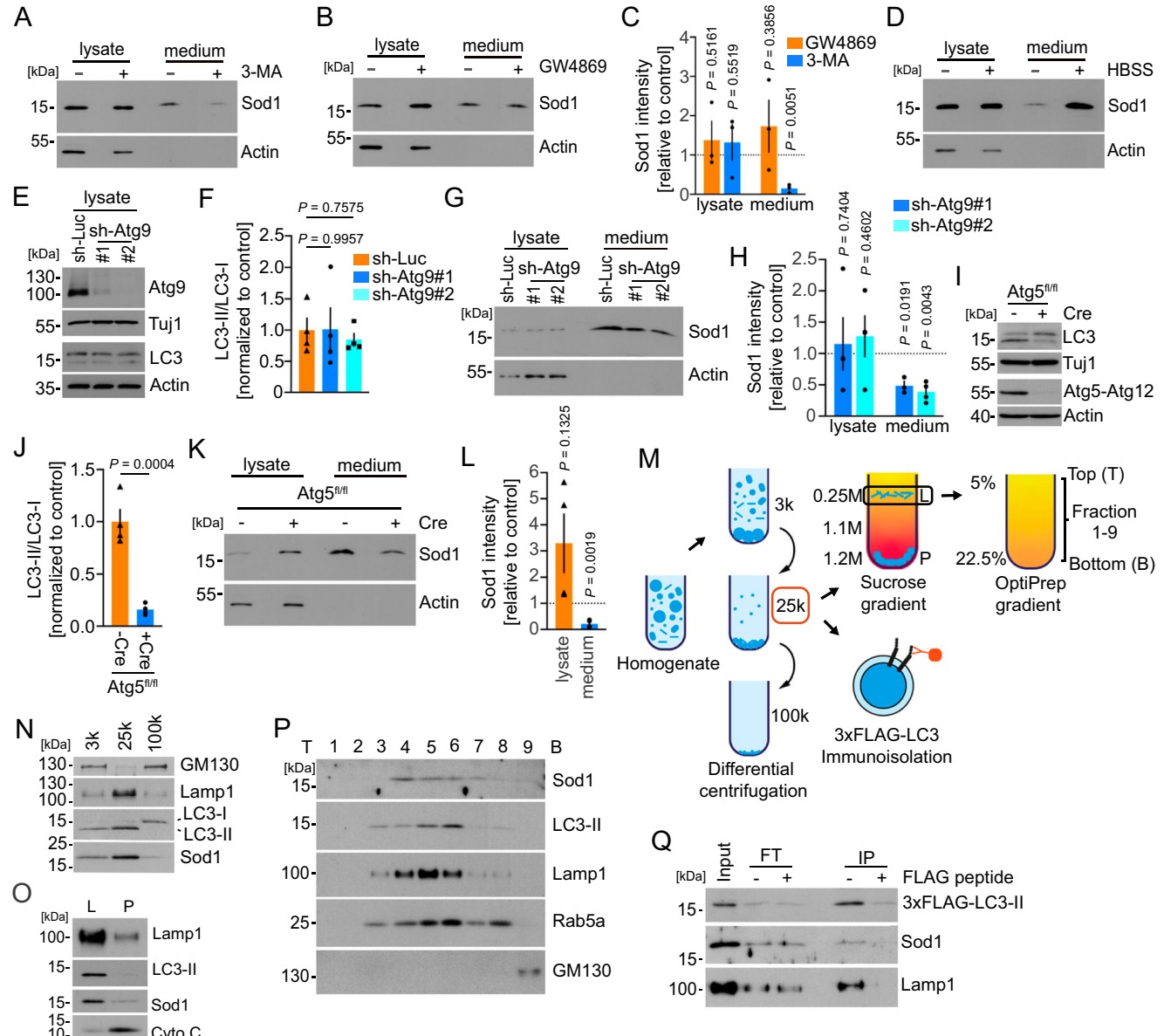

**Fig. 3 | Sod1 is secreted in an autophagy-dependent manner.** Western blots of lysates and media from NSC34 cells treated with 3-MA (**A**), and GW4869 (**B**) for 8 h. **C** Quantification of the Sod1 intensities upon exposure to 3-MA or GW4869. *n* = 3; biological replicates. One-Sample t-test, two-tailed. **D** Western blot of the lysate and medium from NSC34 cells treated for 1 h with HBSS. The images are representative of four biological replicates. **E** Western blot showing similar LC3-II/ LC3-I ratios upon knockdown of Atg9 in MNs (sh-RNA-1= #1; sh-RNA-2 = #2). **F** Quantification of the LC3-II/LC3-I ratios normalized to the control. *n* = 4 biological replicates. Repeated measures one-way ANOVA. Dunnett's multiple comparisons test. **G** Western blots showing reduced Sod1 levels in the medium of MNs upon knockdown of Atg9 in MNs. **H** Quantification of the Sod1 intensities. Lysate *n* = 4, medium *n* = 3 (sh-Atg9#1); Lysate *n* = 4, medium *n* = 4 (sh-Atg9#2); biological replicates. One-sample t-test, two-tailed. **I** Western blot showing a reduced LC3-II/LC3-I ratio in Atg5 depleted MNs. **J** Quantification of the LC3-II/LC3-I ratios normalized to the control.

*n* = 4 biological replicates. t-test, two-tailed. **K** Depletion of Atg5 in MNs results in a reduced secretion of Sod1 as shown by Western blot. **L** Quantification of the Sod1 intensities. *n* = 4 biological replicates. unpaired t-test, two-tailed. **M** Membrane fractionation scheme. **N–P** Western blots showing the distribution of Sod1 and the indicated membrane markers in the different membrane fractions. T, top; B, bottom. The images are representative of two biological replicates. **Q** LC3-positive membranes were immunoisolated and the presence of Sod1 was determined by Western blot. FT flowthrough, IP immunoprecipitation. The images are representative of two biological replicates. Quantification of Sod1 secretion was calculated as the ratio between the amount of Sod1 in the medium and in the lysate. Sod1 levels in lysates were adjusted to Actin. The normalized Sod1 intensities were set to 1 in each experiment Data are mean ± SEM. Source data are provided as a Source Data file.

co-distributed in the L fraction at the boundary between the 0.25 M and 1.1 M sucrose layers (Fig. 3O). Using an OptiPrep gradient, we further fractionated the L fraction and observed a co-fractionation of Sod1 and LC3-II (Fig. 3P). However, Sod1 also co-fractionated with membranes of the endosomal/ lysosomal pathway, as shown by Rab5 and Lamp1 immunoreactivity (Fig. 3P). To confirm the presence of Sod1 in autophagosomes, we expressed 3xFlag-LC3 in NSC34 cells and immunoisolated autophagosomes from the 25k pellet. Subsequently, we analyzed

the input, flowthrough, and IP by Western blot and found that Sod1 co-sedimented with autophagosomes as shown by immunoreactivity for LC3-II (Fig. 3Q). Taken together, these data suggest that a fraction of intracellular Sod1 is directly associated with autophagosomes.

**Sod1 accumulates in autolysosomes in Plekhg5-deficient mice**
The vesicle-like staining pattern of Sod1 in Plekhg5-deficient mice suggests that the accumulated fraction of Sod1 is membrane-bound.

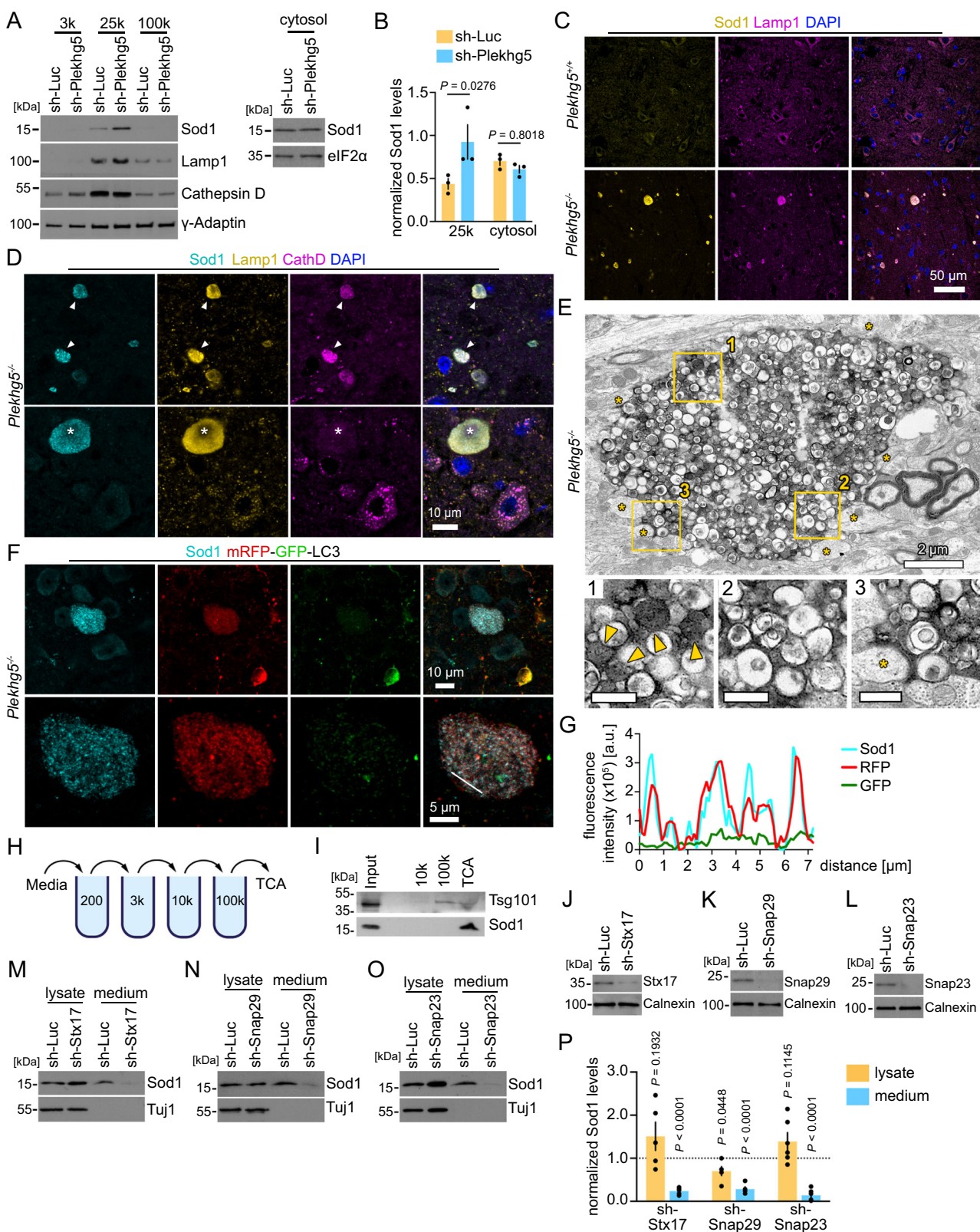

To validate this hypothesis, we performed subcellular fractionation experiments on NSC34 cells depleted of Plekhg5 (Fig. 4A). We separated the cytosol from membranes by a three-step membrane fractionation procedure and detected an increase of the Sod1 levels in the 25k membrane pellet, but not in the cytosolic fraction (Fig. 4A, B).

To characterize the identity of the compartment in which Sod1 accumulates in more detail, we analyzed the Sod1+ vesicle cluster within the spinal cord of Plekhg5-deficient mice by immunohistochemistry using markers for auto- and endolysosomal vesicles. This approach revealed that the majority of Sod1 accumulations stained positive for the late endosomal/ lysosomal marker Lamp1 (Fig. 4C). A subset of the Sod1+ vesicle clusters also stained double positive for Lamp1 and CathD (Fig. 4D). At the ultrastructural level, these accumulations appeared as a densely packed vesicle population which

**Fig. 4 | Sod1 accumulates in autolysosomal structures within the spinal cord of Plekhg5-deficient mice. A** Western blots showing an enrichment of Sod1 in the 25k membrane fraction upon Plekhg5 knockdown in NSC34 cells. **B** Quantification of the Sod1 intensities. The Sod1 intensity of the 25k pellet was normalized to γ-Adaptin, the cytosolic fraction was normalized to eIF2α. *n* = 3 biological replicates. Two-way ANOVA; Sidak's multiple comparison test. **C** Sod1 accumulations in *Plekhg5*$^{-/-}$ mice stained positive for the endosomal/lysosomal marker Lamp1. Co-staining of spinal cord cross sections for Sod1, Lamp1, and DAPI. The images are representative of three biological replicates. **D** A subset of Sod1$^+$Lamp1$^+$ vesicle clusters showed immunoreactivity for the lysosomal marker CathepsinD. Arrow-heads point to CathepsinD$^+$ vesicle clusters. The asterisk labels a CathepsinD$^-$ vesicle cluster. The images are representative of three biological replicates. **E** Electron microscopic micrograph of a vesicle cluster in the spinal cord of *Plekhg5*$^{-/-}$ mice. Postsynaptic spines adjacent to the vesicle cluster are labeled by asterisks. Electron-dense organelles are highlighted by arrowheads. The images are representative of two biological replicates. **F** Co-staining of Sod1 and mRFP-GFP-LC3 in spinal cord cross-sections from *Plekhg5*$^{-/-}$ *CAG:::mRFP-GFP-LC3* mice. The images are representative of three biological replicates. **G** Line scan of the vesicles shown in the lower panel of **F**. **H** Scheme for differential centrifugations of extracellular vesicles. **I** Enrichment of Sod1 in the TCA fraction of medium from NSC34 cells as shown by Western blot. The images are representative of two biological replicates. Western blots showing the knockdown of Stx17 (**J**), Snap29 (**K**), and Snap23 (**L**) upon simultaneous expression of two sh-RNAs in MNs. The images are representative of three biological replicates. Western blots show that the Sod1 secretion is blocked upon knockdown of Stx17 (**M**), Snap29 (**N**), and Snap23 (**O**). **P** Quantification of the Sod1 intensities. sh-Stx17, *n* = 4; sh-Snap29, *n* = 6; sh-Snap23, *n* = 6, biological replicates. One-sample t-test, two-tailed. Quantification of Sod1 secretion was calculated as the ratio between the amount of Sod1 in the medium and in the lysate. Sod1 levels in the lysates were adjusted to Tuj1. The normalized Sod1 intensities were set to 1 in each experiment. Data are mean ± SEM. Source data are provided as a Source Data file.

mostly consisted of single but also double-membraned organelles (Fig. 4E). The vesicles were either electron-dense or filled with a translucent, uncompacted cytoplasmic content (Fig. 4E). The vesicle clusters appeared in direct contact with dendritic spines (Fig. 4E), which further supports the idea that the vesicle accumulations represent swollen presynaptic terminals, as previously reported[17].

We crossed Plekhg5-deficient mice with mRFP-GFP-LC3 reporter mice to examine the presence of the autophagosomal membranes at the Sod1 accumulations (Fig. 4F). This strategy also allows for discrimination between autophagosomes, which appear positive for mRFP and GFP, and autolysosomes, which are only positive for mRFP due to the acidic pH of lysosomes. Whereas GFP was largely absent from the vesicles, the Sod1 vesicle cluster stained positive for mRFP (Fig. 4F, G).

By fusion of the outer autophagosomal membrane with a lysosome, the inner autophagosomal membrane is degraded. Therefore, the subsequent fusion of an autolysosome with the plasma membrane delivers its cargo into the extracellular milieu as a free protein. In this scenario, Sod1 would be released into the media as a free protein, and not within an extracellular vesicle. In agreement with this hypothesis, differential centrifugation of the media revealed that the majority of Sod1 is present as a free protein (Fig. 4H, I). As a control, we used Tsg101, which is present on extracellular vesicles and enriched in the 100k pellet (Fig. 4I). From these data, we conclude that Sod1 is sequestered into an autophagosomal carrier, which subsequently fuses with a lysosome and/or lysosomal related organelle (LRO), thereby generating a secretory autolysosome.

To mechanistically confirm that the secretion of Sod1 requires the fusion between autophagosomes and lysosome, we genetically targeted Stx17 and Snap29 (Fig. 4J, K). While both SNAREs are essential for the fusion between autophagosomes and lysosomes[43,44], Snap29 has also been implicated in the fusion of autophagosomes with the plasma membrane[45]. Based on previous findings, we also targeted Snap23, a SNARE protein involved in lysosomal exocytosis (Fig. 4L)[46,47]. Knockdown of Stx17 and Snap29 in primary MNs blocked the secretion of Sod1, indicating that the Sod1-containing autophagosomal carrier fuses with lysosomes before fusion with the plasma membrane (Fig. 4M, N, P). Knockdown of Snap23 also significantly reduced the release of Sod1 (Fig. 4O, P), which suggests that the fusion with the plasma membrane depends on Snap23. Interestingly, the depletion of Snap29 also resulted in reduced levels of Sod1 in the lysates, which might point to feedback mechanism to compensate for the impaired delivery to lysosomes.

### Sod1 accumulates in Plekhg5-deficient mice due to an impaired secretion of lysosomal-related organelles, but not due to lysosomal dysfunction

We hypothesized that Sod1 might be released by secretory autophagy to prevent its accumulation in autolysosomes due to its resistance against enzymatic cleavage[32]. Therefore, we analyzed the secretion of Sod1 in NSC34 cells upon lysosomal disruption by the V-ATPase inhibitor Bafilomycin A1, which blocks the acidification of lysosomes (Fig. 5A). As a delayed secondary effect, Bafilomycin also blocks the fusion of autophagosomes with lysosomes[48]. To more specifically inhibit lysosomal function, and not fusion between both organelles, we applied a combination of the aspartic protease inhibitor Pepstatin A and the cysteine protease inhibitor E64D (Fig. 5B). Indeed, we detected an enhanced Sod1 secretion upon blockage of lysosomal acidification or inhibition of lysosomal proteolysis, which indicates that Sod1 is secreted in response to lysosomal dysfunction (Fig. 5C). To exclude Sod1 accumulations build-up due to an impaired lysosomal function in Plekhg5-deficient mice, we measured the activity of the lysosomal enzymes β-hexosaminidase, β-galactosidase and β-glucuronidase in spinal cord lysates of Plekhg5-deficient mice, but detected no difference compared to lysates from control animals (Fig. 5D). To confirm that the Lamp1$^+$ vesicle clusters do not represent an accumulation of dysfunctional lysosomes, we stained for the autophagy receptor p62 and Ubiquitin. Both proteins accumulate upon lysosomal dysfunction as previously described[49,50]. The absence of both, p62 and Ubiquitin from Lamp1$^+$ clusters confirms that depletion of Plekhg5 did not result in lysosomal dysfunction (Fig. 5E–G). Therefore, we concluded that the vesicle clusters do not represent dysfunctional autolysosomes, which cause the accumulation of Sod1. Instead, we hypothesized that the Sod1$^+$ Lamp1$^+$ vesicles might represent secretory LROs, which represent cell-type specific secretory organelles[51]. To test this hypothesis, we analyzed the exocytosis of Lamp1$^+$ organelles in cultured MNs. Upon fusion with the plasma membrane, the luminal domain of Lamp1 is exposed to the extracellular milieu, which enables the measurement of the Lamp1 cell surface levels by adding antibodies to the medium of living cells specifically binding to the luminal domain. Using this approach, we detected a reduced Lamp1 staining in Plekhg5-deficient cells (Fig. 5H). Based on previous and recent work suggesting the involvement of Rab26 in endo- and exocytosis, and in the endo/lysosomal system[18,52,53], we hypothesized that Rab26 might regulate the exocytosis of Lamp1$^+$ LROs. Indeed, the expression of Rab26-QL in Plekhg5-depleted MNs restored the plasma membrane levels of Lamp1 (Fig. 5H, I).

In summary, these data show that Sod1 is sequestered into an autophagosomal carrier that fuses with Lamp1$^+$ lysosomalrelated organelles. Fusion of these organelles with the plasma membrane to release Sod1 requires the Plekhg5-mediated activation of Rab26.

### Plekhg5-depletion in SOD1$^{G93A}$ mice accelerates the disease onset but extends the survival

To study the effect of Plekkhg5-depletion on the secretion of disease-relevant mutant ALS-linked SOD1, we crossed Plekhg5-deficient mice with the "low copy" SOD1$^{G93A}$ model of ALS. These SOD1$^{G93A}$ mice show

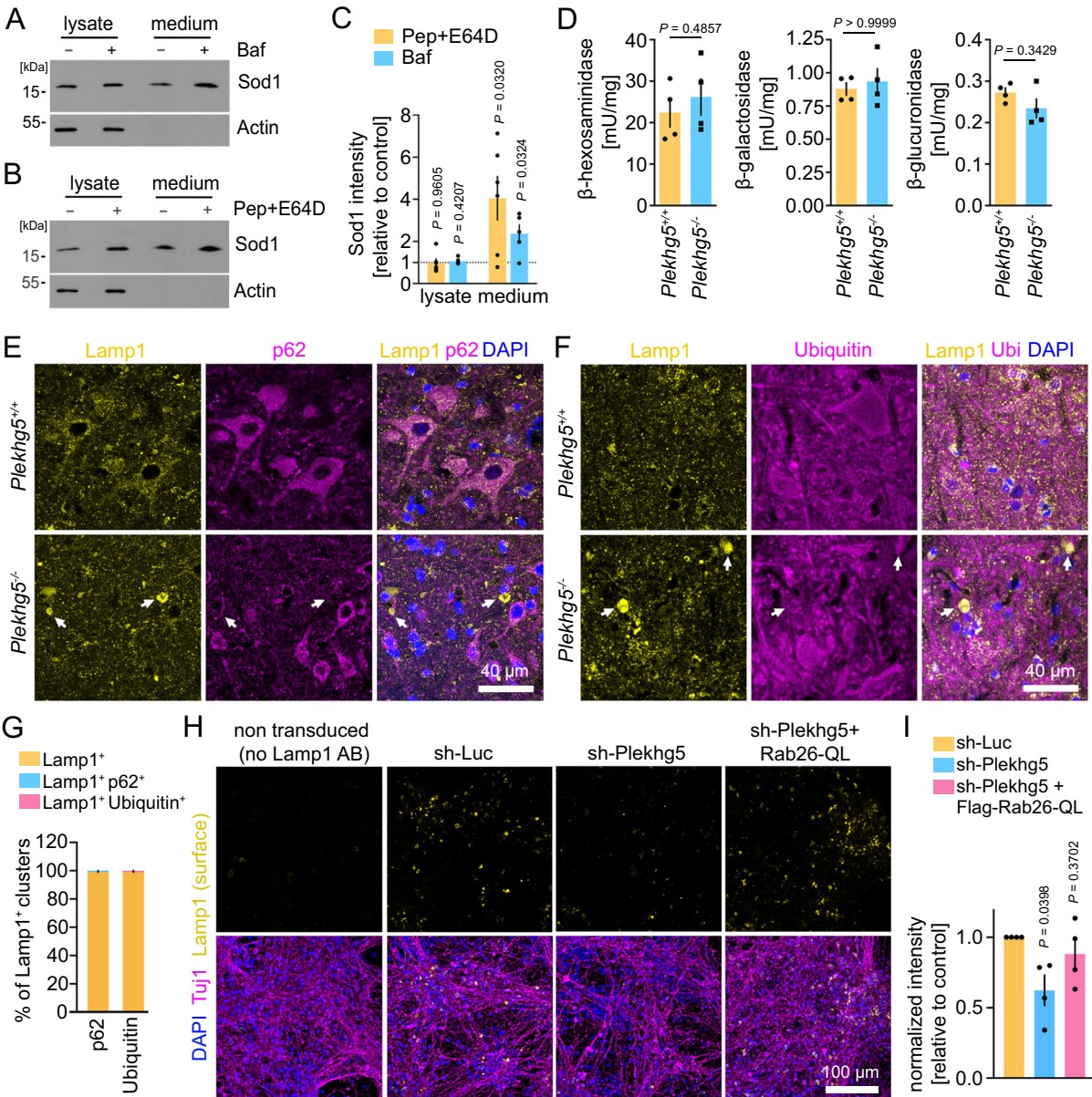

**Fig. 5 | Sod1 accumulates in Plekhg5-deficient mice due to impaired secretion of lysosomal-related organelles, but not due to lysosomal dysfunction.** Elevated Sod1 levels in the medium upon lysosomal dysfunction. Western blots of lysates and media from NSC34 cells treated with Bafilomycin A1 (Baf) (**A**) or Pepstatin A and E64D (**B**) for 8 h. **C** Quantification of the Sod1 intensities. Pep+E64D, $n = 6$; Baf, $n = 5$; biological replicates. One-Sample t-test, two-tailed. Quantification of Sod1 secretion was calculated as the ratio between the amount of Sod1 in the medium and in the lysate Sod1 levels in the lysates were adjusted to Actin. The normalized Sod1 intensities were set to 1 in each experiment. **D** Quantification of the activity of the lysosomal enzymes ß-hexosaminidase, ß-galactosidase, and ß-glucuronidase in spinal cord lysates reveals no difference between *Plekhg5*$^{+/+}$ and *Plekhg5*$^{-/-}$ mice. $n = 4$ biological replicates. Mann–Whitney test, two-tailed. Lamp1 accumulations in *Plekhg5*$^{-/-}$ mice stained negative for p62 (**E**) and Ubiquitin (**F**). Co-staining of spinal cord cross sections from *Plekhg5*$^{-/-}$ and *Plekhg5*$^{+/+}$ mice labeled for Lamp1, p62 and DAPI (**E**) and Lamp1, Ubiquitin and DAPI (**F**). Arrows point to Lamp1$^+$ vesicle clusters. **G** Quantification of the overlap between the immunoreactivity of the Lamp1$^+$ accumulations with p62 or Ubiquitin. Five spinal cord cross sections of three animals were analyzed. $n = 3$ biological replicate. **H** Expression of Flag-Rab26-QL restored the reduced Lamp1 surface levels in MNs upon depletion of Plekhg5. **I** Quantification of the Lamp1 intensity normalized to the number of nuclei. Each dot represents the mean of at least 5 images quantified. $n = 4$ biological replicate. One sample t-test, two-tailed. Data are mean ± SEM. Source data are provided as a Source Data file.

a slow disease progression over several months and survive up to 10 months[54]. At that age, Plekhg5-deficient mice do not show any disease symptoms, but vesicle accumulations are already histologically detectable[17].

To determine any changes in the disease course, we measured the disease onset and the survival of *Plekhg5*$^{+/+}$ *SOD1*$^{G93A}$ and *Plekhg5*$^{-/-}$ *SOD1*$^{G93A}$ mice as previously described (Fig. 6A–E)[55]. We found that depletion of Plekhg5 in *SOD1*$^{G93A}$ mice accelerated the disease onset by 4 weeks (Fig. 6A), but also extended the survival for 4 weeks (Fig. 6B).

In *SOD1*$^{G93A}$ mice, the disease onset became apparent at a mean age of 36 weeks. *Plekhg5* deficiency accelerated the disease onset to a mean age of 32 weeks (Fig. 6C). Remarkably, the median age of survival was extended from 44 weeks in *SOD1*$^{G93A}$ mice to 48 weeks in *SOD1*$^{G93A}$ *Plekhg5*$^{-/-}$ *SOD1*$^{G93A}$ mice (Fig. 6D). As a result, the average disease duration was extended from 8 to 16 weeks (Fig. 6E).

Next, we asked whether Plekhg5 had any effects on the accumulation of the mutant SOD1$^{G93A}$ protein. At the disease onset, we already observed a striking clustering of SOD1$^{G93A}$ in *Plekhg5*$^{-/-}$ *SOD1*$^{G93A}$ mice,

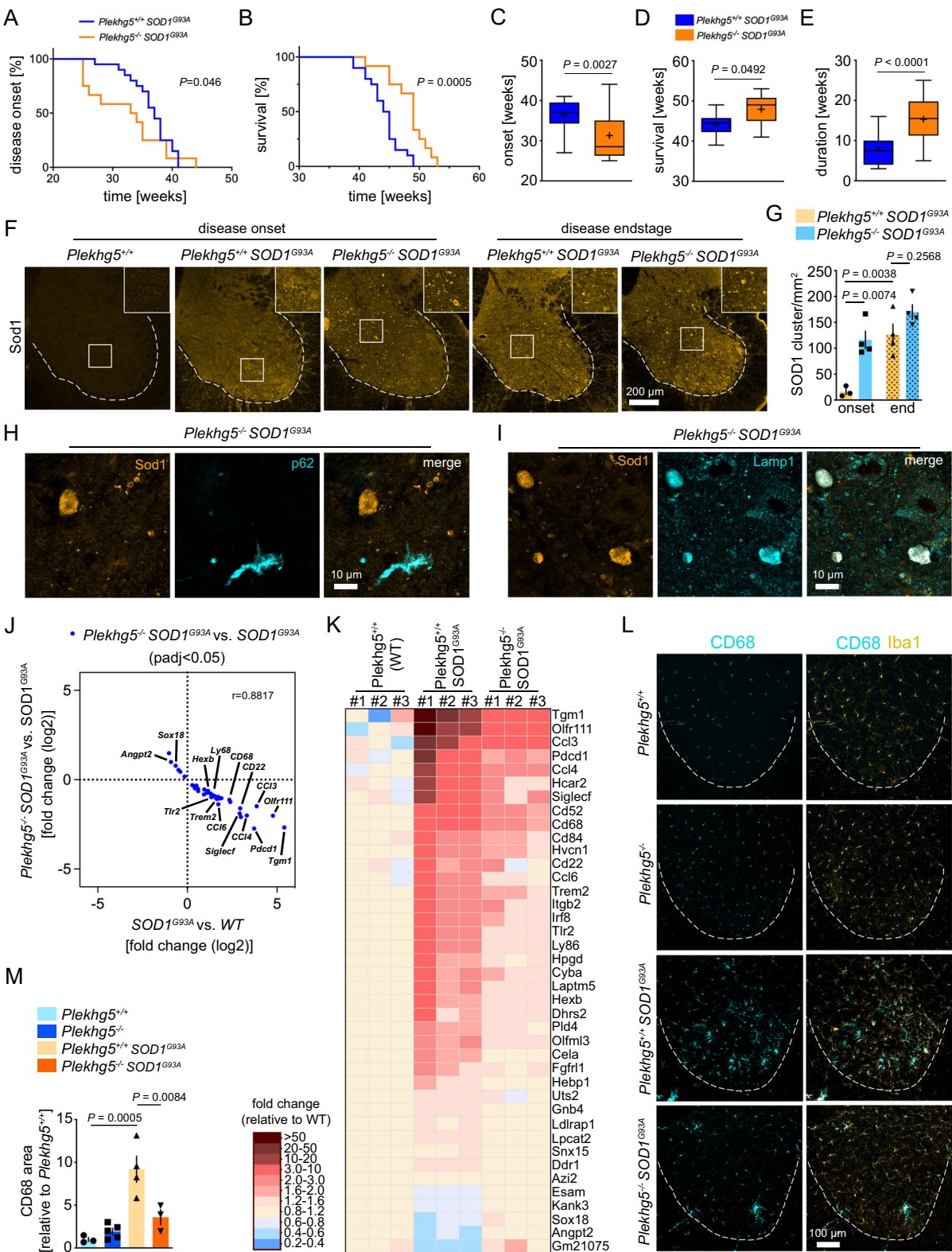

whereas *Plekhg5*[+/+] *SOD1*[G93A] animals only displayed a sparse accumulation of SOD1[G93A] (Fig. 6F). However, with disease progression, the number of SOD1[G93A] clusters increased in *Plekhg5*[+/+] *SOD1*[G93A] mice (Fig. 6G). In contrast, the number of SOD1[G93A] clusters in *Plekhg5*[−/−] *SOD1*[G93A] did not increase anymore. At the endstage, we detected comparable numbers of SOD1[G93A] accumulations in both genotypes (Fig. 6F, G). These data suggest that the mutant, ALS-linked SOD1[G93A] is

affected in a similar way by depletion of Plekhg5 than the endogenous wildtype Sod1.

A major pathological hallmark in SOD1[G93A] mice is the presence of p62[+] aggregates caused by impaired global proteostasis[55]. In line with our finding that p62 and Ubiquitin were absent from the Lamp1[+]Sod1[+] vesicle cluster in Plekhg5-deficient mice, the depletion of Plekhg5 in SOD1[G93A] mice did not affect the number of p62 aggregates within the

**Fig. 6 | Depletion of Plekhg5 in *SOD1^G93A* mice prepones the disease onset but extends the survival.** Kaplan–Meier plots showing the onset of weight loss (**A**) and survival (**B**) of *Plekhg5^-/- Sod1^G93A* mice compared to *Plekhg5^+/+ Sod1^G93A*. Log-rank (Mantel Cox) test. Average disease onset (**C**), survival (**D**), and duration (**E**). **A**, **C** *Plekhg5^+/+ Sod1^G93A*, n = 20. *Plekhg5^-/- Sod1^G93A*, n = 16. **B**, **D**, **E** *Plekhg5^+/+ Sod1^G93A*, n = 20. *Plekhg5^-/- Sod1^G93A*, n = 12; biological replicates. Two-tailed t-test. Box bounds are defined by min to max. Whiskers represent data points within 1.5 IQR. Lines and crosses denote the median and mean. **F** Sod1 staining of spinal cord cross sections revealed an accelerated accumulation of Sod1 in *Sod1^G93A* mice upon depletion of *Plekhg5*. **G** Quantification of Sod1 accumulations. Onset: *Plekhg5^+/+ Sod1^G93A*, n = 3; *Plekhg5^-/- Sod1^G93A*, n = 4; biological replicates. Endstage: *Plekhg5^+/+ Sod1^G93A*, n = 4; *Plekhg5^-/- Sod1^G93A*, n = 4; biological replicates. One-way ANOVA; Tukey's Multiple Comparisons. Sod1 accumulations in *Plekhg5^-/-* Sod1^G93A mice stain negative for p62 (**H**) but positive for Lamp1 (**I**). The images are representative of three biological replicates. **J** Scatterplot showing the magnitude of change (fold change, log2) of the transcripts significantly altered in SOD1^G93A vs. wildtype (WT) mice and significantly altered in SOD1^G93A mice vs. Plekhg5^-/- SOD1^G93A. n = 3 biological replicates. Pearson correlation, r = −0.9390. **K** Heat Map showing the relative expression levels of transcripts significantly altered in SOD1^G93A vs. Plekhg5^+/+ (WT) mice and significantly altered in SOD1^G93A mice vs. Plekhg5^-/- SOD1^G93A. The expression levels are shown as fold change of the normalized read counts adjusted to the mean levels of Plekhg5^+/+ (WT) mice. **L** Staining for Iba1 and CD68 in spinal cord cross-sections revealed a reduced microglial neuroinflammation in *Plekhg5^-/- Sod1^G93A* mice. **M** Quantification of the CD68 immunoreactivity area. *Plekhg5^+/+*, n = 3; *Plekhg5^-/-*, n = 5; *Plekhg5^+/+ Sod1^G93A*, n = 4; *Plekhg5^-/- Sod1^G93A*, n = 3; biological replicates. One-way ANOVA; Tukey's Multiple Comparisons. Data are mean ± SEM. Source data are provided as a Source Data file.

spinal cord (Supplementary Fig. 2). Furthermore, p62 was not detectable at Sod1 accumulations in *Plekhg5^-/-* SOD1^G93A mice (Fig. 6H), but the Sod1 accumulations stained positive for Lamp1 (Fig. 6I). This data set confirms that the Sod1 accumulations caused by depletion of Plekhg5 are distinct from p62^+ protein aggregates.

To characterize the effect of Plekhg5 depletion on the *SOD1^G93A* model in more detail, we analyzed MN survival by quantifying the number of ChAT-positive neurons in the ventral horn of the lumbar spinal cord and the denervation and morphology of the NMJs (Supplementary Fig. 3A–E). Whereas no change in the MN number was detectable at disease onset, a loss of MNs became apparent at the endstage (Supplementary Fig. 3A, B). In *Plekhg5^-/- SOD1^G93A* mice the survival of MNs was slightly but significantly improved compared to *Plekhg5^+/+ SOD1^G93A* mice (Supplementary Fig. 3B). Next, we examined the effect of Plekhg5 depletion on NMJ degeneration (Supplementary Fig. 3C–E). We measured the number of denervated NMJs and the number of swollen, "balloon-like" NMJs, a characteristic phenotype of Plekhg5-deficient mice (Supplementary Fig. 3D, E)[17]. We found a progressive denervation of the gastrocnemius muscle in *SOD1^G93A* during the disease course (Supplementary Fig. 3D). At the disease endstage, *Plekhg5^-/- SOD1^G93A* animals retained a higher number of innervated NMJs than their *Plekhg5^+/+ SOD1^G93A* counterparts. When we analyzed the morphology of the NMJs in more detail, we detected an age-dependent increase of balloon-like NMJs in *Plekhg5^-/-* animals (Supplementary Fig. 3E). In addition, *SOD1^G93A* mice deficient for Plekhg5 also acquired this phenotype, but surprisingly to a significantly lesser extend as compared to *Plekhg5^-/-* animals (Supplementary Fig. 3E). In summary, Plekhg5 depletion in *SOD1^G93A* mice preserves NMJ innervation at late disease stages. On the other hand, NMJs denervation due to SOD1^G93A expression also results in pruning of "balloon-like" terminals.

## Plekhg5-depletion in SOD1^G93A ameliorates microglial neuroinflammation

Previous work demonstrated that secreted, mutant ALS-linked SOD1 protein acts as danger-associated molecular patterns (DAMPs) and binds as a ligand to TLR2 receptors driving microglia activation[56,57]. Therefore, we hypothesized that an impaired SOD1 secretion leads to an attenuated microglial neuroinflammation. During disease progression, microglia from *SOD1^G93A* mice acquire a unique gene-expression signature[58]. To study whether the depletion of Plekhg5 in *SOD1^G93A* mice caused any changes in this signature, we utilized an unbiased approach and performed RNAseq of total spinal cord RNA (Fig. 6J).

We compared the transcripts differentially expressed in *SOD1^G93A* mice versus wildtype mice, with transcripts differentially expressed in *Plekhg5^-/- SOD1^G93A* mice versus SOD1^G93A mice (Fig. 6J). We found a total number of 40 transcripts that were differently expressed in both data sets (Fig. 6J, K). Strikingly, all 34 transcripts, which were upregulated in SOD1^G93A mice, were downregulated in *Plekhg5^-/- SOD1^G93A* mice (Fig. 6J, K). Conversely, the expression of all 6 upregulated transcripts in *SOD1^G93A* mice was downregulated upon Plekhg5 deficiency

(Fig. 6J, K). A comparison with the microglial-specific ALS signature revealed that 18 of the 34 downregulated transcripts overlapped with genes whose expression is upregulated during disease progression[58]. To confirm the RNAseq data, we analyzed the changes in the levels of several transcripts by qPCR (Supplementary Fig. 3F). In addition, we confirmed a reduced microglial activation in *SOD1^G93A* mice with Plekhg5 depletion by staining for the microglial markers Iba1 and CD68 (Fig. 6L). Whereas no microglia activation could be detected in wildtype and Plekhg5^-/- animals, we observed a significant increase in the CD68 immunoreactivity in *Plekhg5^+/+ SOD1^G93A* mice, which was significantly reduced in *Plekhg5^-/- SOD1^G93A* mice (Fig. 6L, M). Taken together, these data suggest that the impaired SOD1^G93A secretion upon Plekhg5 depletion is also reflected in changes of the neuroinflammatory signature of *SOD1^G93A* mice.

## The secretion of ALS-linked SOD1 is impaired in human iPSC derived MNs

To assess whether PLEKHG5 also mediates the UPS of human wildtype SOD1 and most importantly ALS-linked SOD1, we examined the levels of SOD1 in lysates and media of hiPSCs-derived MNs. Previously, two isoforms of human PLEKHG5 have been described, which differ in their N- and C-terminus[24]. In hiPSC-derived MNs, we detected both isoforms, with the shorter isoform 1 expressed at higher levels (Fig. 7A). Knockdown of both PLEKHG5 isoforms in hiPSCs derived MNs from heathy donors resulted in reduced SOD1 levels in the cell culture medium (Fig. 7B, C) (Supplementary Fig. 4A), which confirms our data obtained with primary mouse MNs and NSC34 cells (Fig. 2B–E). In line with our observations in Plekhg5-deficient mice, we also detected deformed axon terminals with an enrichment of SOD1 in hiPSC-derived MNs upon knockdown of PLEKHG5 (Fig. 7D, E).

Previous work indicated that the secretion of mutant ALS-linked SOD1^G93A is reduced compared to its wildtype counterpart upon overexpression in Hela cells[14]. We wondered whether differences in the secretion of ALS-linked human SOD1 expressed at physiological levels are detectable in human MNs. Therefore, we utilized two previously characterized iPSC lines derived from ALS patients, either harboring a heterozygous R115G mutation or a homozygous D90A mutation in the *SOD1* gene[59,60]. While the R115G mutation in SOD1 gives rise to structurally unstable proteins, the SOD1^D90A mutant protein is structurally stable and has enzymatic activity that is as high as wildtype SOD1[61]. We compared the secretion of these two mutant SOD1 species to the secretion of wildtype SOD1, which was analyzed in two lines derived from healthy donors and an isogenic line with a corrected *SOD1^D90A/D90A* allele as an additional control (Fig. 7F). To examine the secretion of WT and ALS-linked SOD1 in MNs, we differentiated the iPSC lines as previously described[60], and confirmed similar differentiation efficiencies into MNs by Islet and Tuj1 staining (Supplementary Fig. 4B, C).

Notably, we found reduced levels of ALS-linked SOD1 in the cell culture medium, with a more pronounced effect in the case of

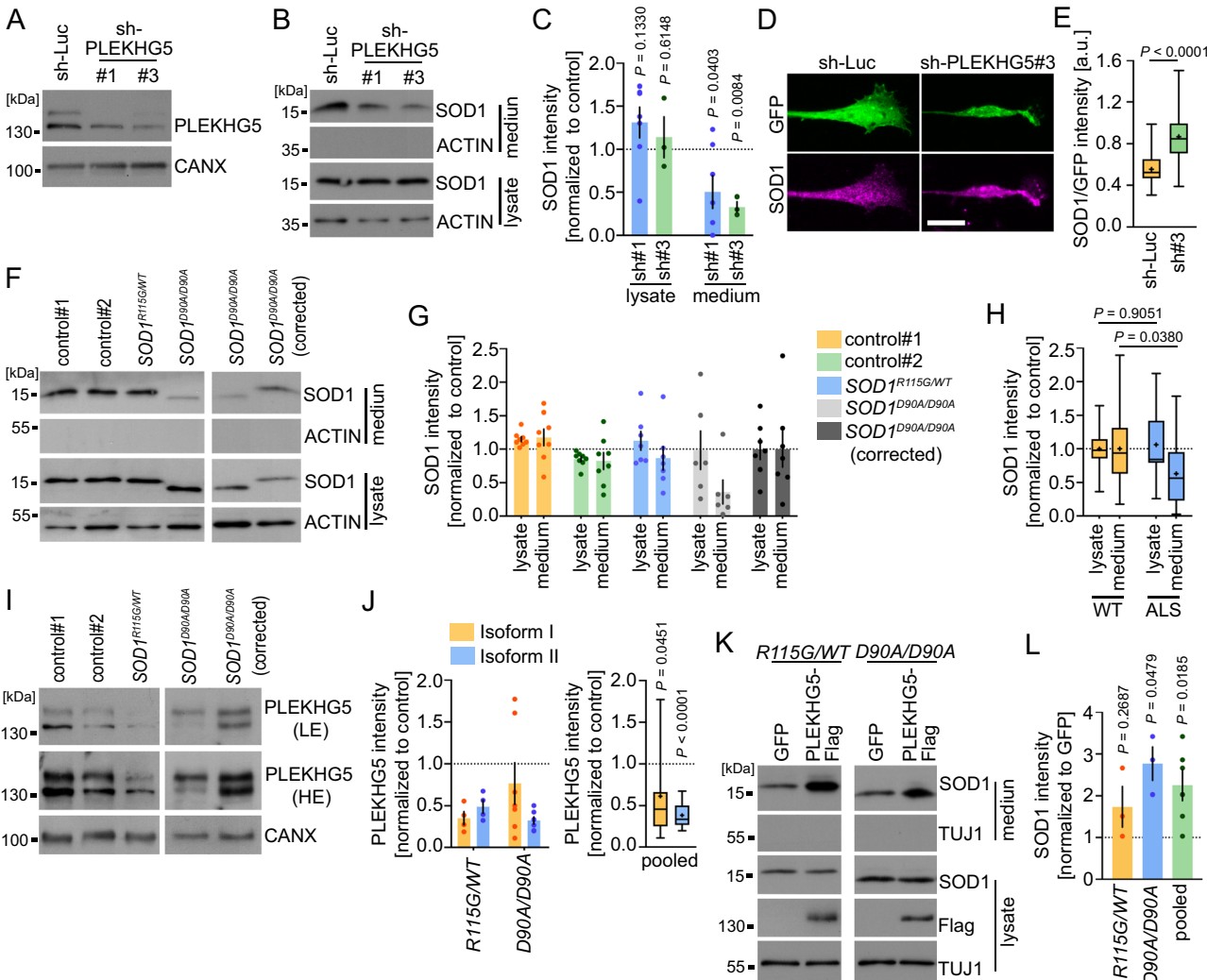

**Fig. 7 | PLEKHG5 drives the secretion of mutant ALS-linked SOD1 in human iPSC-derived MNs. A** Western blot showing the knockdown of PLEKHG5 in hiPSC-derived MNs (sh-RNA-1 = #1; sh-RNA-3 = #3). The images are representative of two biological replicates. **B** Western blot showing reduced SOD1 levels in the medium of PLEKHG5-depleted MNs. **C** Western blot quantification of the SOD1 intensities. One sample t-test, two-tailed. sh#1, $n = 6$; sh#3, $n = 3$; biological replicates. **D** Immunoreactivity of SOD1 in axon terminals of control or PLEKH5-depleted hiPSC-derived MNs. Scale bar: 10 μm. **E** Quantification of the SOD1 intensity normalized to GFP. sh-Luc, $n = 48$; sh-PLEKHG5, $n = 52$. Three biological replicates. Two-tailed test, unpaired. **F** Western blots showing reduced SOD1 levels in the medium of iPSC-derived MNs with mutant ALS-linked SOD1. **G, H** Western blot quantification of the SOD1 intensity. **G** Individual data: control#1, $n = 8$; control#2, $n = 8$; $SOD1^{R115G/WT}$, $n = 7$; $SOD1^{D90A/D90A}$, $n = 6$; $SOD1^{D90A/D90A}$ (corrected); biological replicates. **H** Pooled data: WT, $n = 23$; ALS, $n = 13$; biological replicates. Two-way ANOVA, Sidak's multiple comparison test. **I** Western blots showing reduced

PLEKHG5 levels in the lysates of $SOD1^{D90A/D90A}$ and $SOD1^{R115G/WT}$ MNs. **J** Western blot quantification of the PLEKHG5 intensity relative to CANX. $SOD1^{R115G/WT}$, $n = 4$; $SOD1^{D90A/D90A}$, $n = 7$. Pooled, $n = 11$; biological replicates. One sample t-test, two-tailed. LE low exposure, HE high exposure. **K** Expression of Flag-Plekhg5 (Isoform 1) caused an increase in the media of mutant ALS-linked SOD1 as shown by Western blot. **L** Western blot quantification of the SOD1 intensities. $SOD1^{R115G/WT}$, $n = 3$; $SOD1^{D90A/D90A}$, $n = 3$; Pooled, $n = 6$; biological replicates. One sample t-test, two-tailed. Quantification of SOD1 secretion was calculated as the ratio between the amount of SOD1 in the medium and in the lysate. The SOD1 levels in the lysates were adjusted to CANX. The SOD1 or PLEKHG5 levels of $SOD1^{R115G/WT}$ MNs were normalized to the mean value of both controls. The intensities of $SOD1^{D90A/D90A}$ MNs were normalized to their corrected counterpart. The normalized SOD1 intensities were set to 1 in each experiment. Data are mean ± SEM. Box bounds are defined by min to max. Whiskers represent data points within 1.5 IQR. Lines and crosses denote the median and mean. Source data are provided as a Source Data file.

$SOD1^{D90A}$ (Fig. 7F–H). As described previously, we observed slightly increased mobility of $SOD1^{D90A}$ in the SDS-PAGE[62].

In order to establish a potential link between the reduced secretion of mutant SOD1 and PLEKHG5, we analyzed the expression levels of PLEKHG5 in both ALS lines by Western blot (Fig. 7I) and detected a significantly lower expression of both PLEKHG5 isoforms in the SOD1-ALS lines (Fig. 7J). In line with this observation, a downregulation of *PLEKHG5* transcripts has recently also been identified in post-mortem spinal cord samples from ALS patients[63]. In contrast to our observation in hiPSC-derived MNs, we did not detect a reduced Plekhg5 expression in spinal cord lysates of 8 months old $SOD1^{G93A}$ mice (Supplementary Fig. 4D, E). However, if the downregulation of Plekhg5 is MN-specific,

the differences in the expression levels might be too small for the detection by Western blot. To examine whether the reduced expression of PLEKHG5 is responsible for the impaired secretion of ALS-linked SOD1, we expressed Flag-tagged PLEKHG5 Isoform 1 in both ALS lines and found a significant increase in the SOD1 medium levels (Fig. 7K, L). Collectively, these data show that PLEKHG5 regulates the release of mutant ALS-linked SOD1.

## Discussion

In summary, our data demonstrate that Plekhg5 regulates the UPS of Sod1 in a Rab26-dependent manner. Depletion of Plekhg5 resulted in intracellular presynaptic accumulations of Sod1 within LROs.

Intriguingly, Sod1 did not accumulate due to a general impairment of the global proteostasis, suggesting a specific route for Sod1 to facilitate its release to the extracellular milieu.

Mechanistically, we show that Sod1 is sequestered into an autophagosomal carrier, which transits through the endo/lysosomal pathway, leading to the fusion with LROs. In the absence of Plekhg5, Sod1 accumulates in such LROs. Due to unaltered lysosomal enzyme activity in Plekhg5-deficient mice and the lack of p62 and Ubiquitin from the Lamp1+/Sod1+ vesicle clusters, we can exclude lysosomal dysfunction in Plekhg5-deficient mice. Furthermore, LROs localized in the periphery show a lower pH than conventional lysosomes and possess a limited degradative capacity, if any at all[51]. In addition, recent work suggested that distal axons lack genuine lysosomes with proteolytic capacity[64]. Therefore, we conclude that the membrane carrier in which Sod1 accumulates upon depletion of Plekhg5 represents secretory Lamp1+ LROs rather than a conventional lysosome/autolysosome. However, co-staining of the Lamp1+/Sod1+ vesicle clusters with Cathepsin D and loss of the GFP signal from the mRFP-GFP-LC3 reporter, demonstrate the acidification of these organelles, at least to a certain degree. By fusion of the outer autophagosomal membrane with a lysosome, the inner autophagosomal membrane is degraded. Therefore, the subsequent fusion of an autolysosome with the plasma membrane delivers its cargo into the extracellular milieu as a free protein. In line with this, we found that Sod1 is rather secreted as a free protein but not in an extracellular vesicle. As recently shown, this contrasts TDP43 and Tau, which are detectable in plasma extracellular vesicles, providing a promising non-invasive biomarker for frontotemporal dementia and ALS[65].

In Plekhg5-deficient mice, Sod1 predominantly accumulated in axons terminating within the spinal cord and MN terminals within muscle. The restriction of this phenotype to the axon requires the compartmentalized action of Plekhg5, which is in agreement with previous work showing that Plekhg5 drives the autophagy of SVs at MN terminals by activating Rab26[17]. The data presented here indicate that Rab26 is also involved in the exocytosis of synaptic LROs. Recently, the proteome of Rab26 organelles has been determined, showing a striking enrichment of SV proteins, followed by endosomal/lysosomal markers[52]. Furthermore, Rab26-containing vesicles displayed many SNAREs proteins, predominantly involved in endosome fusion and exocytosis. Such a signature fits our hypothesis with Rab26 functioning predominantly in the presynaptic compartment involved in regulating endo/lysosomal pathways.

To address if Plekhg5-mediated Sod1 secretion is also relevant for the disease course of ALS-linked Sod1 mutations, we generated Plekhg5⁻/⁻ SOD1^G93A mice. Although Plekhg5⁻/⁻ SOD1^G93A mice showed a markedly accelerated accumulation of SOD1^G93A vesicles and a preponed disease onset, the survival of Plekhg5 depleted SOD1^G93A mice was extended compared to their Plekhg5^+/+ SOD1^G93A counterparts. Interestingly, a similar phenotype was observed in SOD1^G93A mice with MN-specific depletion of autophagy[66]. Notably, these animals displayed an attenuated microglia activation, which indicates that some autophagy-dependent processes in MNs drive microglial neuroinflammation in a non-autonomous manner. It is tempting to speculate that the underlying mechanism involves secretory autophagy, which mediates the release of mutant SOD1. Once released, the mutant SOD1 protein acts as a DAMP and binds to TLR2 receptors on microglia[56,57]. Thus, an impaired secretion of SOD1 causes less microglia activation, attenuating neuroinflammation. Our findings add further weight to the notion that MN degeneration in ALS not only involves cell-intrinsic mechanisms but also depends on non-cell autonomous mechanisms by other cell types[67]. In addition, our data support the idea that unconventional pathways for protein secretion, such as secretory autophagy, are critically involved in the pathophysiology of neurogenerative diseases, which contribute to non-cell autonomous mechanisms of neurodegeneration. Recent studies showed that

lysosomal exocytosis releases pathogenic alpha-Synuclein species, that the secretion of wildtype and mutant Tau depends on secretory autophagy, and that the mutant Huntington is secreted via unconventional secretory pathways[11,13,47]. Secretion of such pathogenic proteins triggers microglial neuroinflammation, driving the non-cell autonomous neurodegeneration. Furthermore, pharmacological inhibition of PIKFYVE kinase activates a UPS pathway involving exocytosis of aggregation-prone proteins[68].

Previous work demonstrated a trans-synaptic propagation of prion-like misfolded protein SOD1^G85R [69], which is in line with our results indicating a marked secretion of Sod1 from axons. It's tempting to speculate that the propagation of Sod1 in the sense of prion-like spreading is also hampered by the depletion of Plekhg5, which might ameliorate disease progression in a non-cell autonomous manner. On the other hand, our data suggest that accumulations within the cell might be intrinsically detrimental, leading to the accelerated disease onset of Plekhg5⁻/⁻ SOD1^G93A.

To explore the effect of ALS-linked mutations on the secretion of SOD1, we utilized human iPSCs derived from ALS patients carrying SOD1 mutations. We investigated MNs derived from iPSCs carrying a heterozygous R115G mutation and a second iPSC line with a homozygous D90A mutation. In both lines, we observed reduced levels of SOD1 in the medium, which coincided with reduced PLEKHG5 protein levels. Lentiviral expression of PLEKHG5 in the SOD1-ALS MNs caused an increase in the SOD1 levels in media, indicating that PLEKHG5 also triggers the UPS of ALS-linked SOD1. The convergence of SOD1 and PLEKHG5 to a common pathophysiological mechanism is of particular interest since both proteins have previously been linked to different forms of MND. The diversity of gene mutations causative for fALS makes it challenging to identify common disease mechanisms. Thus, a pathogenetic mechanism directly affecting at least two MND-associated proteins provides valuable insight into how different, "unrelated" proteins converge into a common pathway. Interestingly, SOD1-ALS is characterized by distinct phenotypic and cellular hallmarks in comparison to typical ALS. In contrast to fALS associated with variants in e.g. C9ORF72, FUS, or TARDBP, there is hardly any cognitive impairment in SOD1-associated ALS[70]. Furthermore, patients with SOD1-ALS often show a lower MN predominant phenotype, with more frequent limb onset compared to typical ALS[71]. TDP-43 protein inclusions, which are the histopathological hallmark in most fALS and sALS cases, are absent in SOD1-ALS patients[72,73]. Vice versa, SOD1 aggregates are hardly detectable in sALS and fALS cases, that are not linked to variants in SOD1[74]. This implies different pathophysiological mechanisms leading to the loss of MNs in SOD1-ALS. Based on our data, it's tempting to speculate that PLEKHG5 is involved in this pathway regulating the burden of intracellular SOD1. However, so far, no variants of PLEKHG5 have been described in SOD1-ALS. Nevertheless, a reduced expression of PLEKHG5 has been identified in spinal cord samples from ALS patients by a recent multiomic and machine learning approach[63], suggesting that the PLEKHG5 levels might be affected by other pathophysiological mechanisms.

Given the increasing number of PLEKHG5 mutations that have been linked to different forms of MND, PLKEKHG5 emerges as one of the key players in preserving the function of MNs and the peripheral nervous system.

## Methods

### Animals and ethical approval

All animal housing and experiments were performed in accordance with the regulations on animal protection of the German federal law, the Association for Assessment and Accreditation of Laboratory Animal Care and the University clinic Wuerzburg guidelines, in agreement with and under the control of the local veterinary authority. Mice were housed in the animal facility of the Institute of Clinical Neurobiology at the University Hospital of Wuerzburg. Mice were kept at 12/12 h light/dark conditions with constant temperature and humidity. Food

and water were provided ad libitum. Food was provided as hybrid pallets from Altromin (1318) during breeding and (1324) during housing and experiments.

All mice strains were kept on a C57BL/6 background. Mice sex was not considered and were equally distributed in the experiments. The original mice strains used in this study are listed in supplementary table 2. The primer sequences for genotyping are listed in Supplementary Table 6. To assess the phenotype of SOD1[G93A] mice, we used previously described criteria[17]. We defined the disease onset as the beginning of weight loss after reaching the peak weight and the survival as the age at which disease-associated paralysis prevented a mouse from righting itself after being placed on its side or reaching a weight-loss of >20%.

## Immunohistochemistry for muscles
The tibialis anterior and gastrocnemius muscles were dissected, and muscle fibers were separated into continuous fiber strands and post-fixed for 2 h in 4% paraformaldehyde (PFA). Muscle fibers were washed three times with PBS, followed by 0.1 M glycin for 30 min, ammonium acetate puffer for 30 min, and incubated in 10% donkey serum with 0.3% Triton X-100 overnight at 4 °C. Three subsequent washing steps with TBS-T containing 0.1% Triton X-100 were performed, and the samples were incubated with primary antibodies in a blocking solution for 2-3 days at 4 °C. The samples were washed three times with TBS-T, incubated with secondary antibodies at RT for 2 h, thoroughly washed with PBS and mounted with FlourSave (Merck, 345789). To visualize postsynaptic Acetylcholine receptors, α-bungarotoxin conjugated to Alexa-488 from Invitrogen (B13422) was used.

## Immunohistochemistry for spinal cord sections
Mice were deeply anesthetized and trans-cardially perfused with 4% PFA. Spinal cords were removed after perfusion, post-fixed for 2 h with 4% PFA and sectional cut with 40 µm thickness at a Leica VT1000S Vibratome. Free-floating spinal cord sections were washed three times with PBS, followed by a 15 min incubation in 0.1 M glycine. Subsequently, the sections were incubated for 30 min in ammonium acetate puffer, washed two times with PBS, and incubated in 10% donkey serum with 0.3% Triton X-100 for 2 h at RT. Three subsequent washing steps with TBS-T containing 0.1% Triton X-100 were performed, and the samples were incubated with primary antibodies in a blocking solution for 2-3 days at 4 °C. Samples were washed three times, and incubated with secondary antibodies at RT for 2 h. After thoroughly washing with PBS, samples were mounted with FlourSave. Acetylcholine receptors were stained with α-bungarotoxin (Invitrogen, B13422) coupled to a 488 fluorophore. Alexa-488-, DyLight-550-, Cy3-, and Cy5-conjugated secondary antibodies were obtained from Jackson Immuno-Research Laboratories. DAPI (Sigma-Aldrich, D9542-5MG) was used for nucleic acid staining. Primary antibodies used for immunohistochemistry are listed in Supplementary Table 4.

## Immunocytochemistry
hiPSC-derived MNs were washed once with PBS, fixed with 4% PFA for 15 min at RT and washed with 0.1% Saponin in TBS-T for 10 min. Subsequently, the cells were washed once with TBS-T and blocked with 10% horse serum in TBS-T for 1 h. The samples were incubated in primary antibody solution containing TBS-T overnight at 4 °C, washed 3 times with PBS and incubated with secondary antibody solution in PBS for 1 h. Cells were washed three times with PBS and mounted in FlourSave.

## Electron microscopy
Mice were trans-cardially perfused, slightly modified according to Forssmann et al.[75]. Briefly, mice blood was rinsed for 2 min and afterwards fixated with 3% PFA, 3% glutaraldehyde (Serva, 23114.01), 0.5% picric acid (Sigma Aldrich, 197378) in 0.1 M sodium phosphate buffer (PB), pH 7.2 for 10 min. Spinal cord segments were post-fixed 1-2 h at

4 °C and osmicated with 2% OsO4 (VWR, 7436.1) for 2.5 h at 4 °C. Samples were embedded in Araldite and cut into 60–80 nm sections, stained with uranyl acetate for 40 min and lead citrate for 8 min. Sections were imaged at a transmission electron microscope (JEOL JEM-2100) at 200 kV, and images were recorded with a TVIPS F416 camera.

## SIM microscopy
Samples were processed identically to spinal cord immunohistochemistry samples. Additionally, Tetraspecks (Invitrogen, T7280) were applied on the new objects slide for chromatic aberration correction. Samples were scanned at a SIM Microscope (Zeiss Elyra S.1 SIM), reconstructed, and channels aligned. SIM stacked images were analyzed via deepflash2. Briefly, new models were generated for SOD1 positive ring-like structures and manually controlled. The total fluorescence intensity of Lamp1 signal was quantified at overlapping ring masks and inside of each ring, respectively.

## Fluorescence signal quantification
Signal quantification was performed by utilizing the ImageJ application in combination with the deepflash2 analysis tool[76]. Five images were manually analyzed by experts, and corresponding masks used for model training. The resulting masks were particle analyzed with ImageJ, quantified, and normalized to the gray matter size of the corresponding spinal cord section.

The Imaris software (7.7×) was used to reconstruct confocal z-stacks as 3D images. The background was filtered by a manually set threshold and quality filters to render fully connected axons.

## Primary mouse MN culture
Murine embryonic spinal MNs were isolated and cultured as previously described[77]. Briefly, after dissection of the ventrolateral part of E13 embryos, spinal cord tissues were incubated for 15 min in 0.05% trypsin in Hank's balanced salt solution (HBSS). Cells were triturated and incubated in Neurobasal medium, supplemented with 100 µg/mL Penicillin/Streptomycin/Glutamax (Gibco, 10378016) on Nunclon plates (Thermo Fisher Scientific, 150350) pre-coated with antibodies against the p75 NGF receptor (MLR2, kind gift of Robert Rush, Flinders University, Adelaide, Australia) for 45 min. Plates were washed with Neurobasal medium, and the remaining MNs were recovered from the plate with depolarization solution (0.8% NaCl, 35 mM KCl and 2 mM CaCl2) and collected in full medium (2% horse serum, 1× B-27 in Neurobasal medium with Glutamax). 2–3 spinal cord tissues were collected and pooled for one experimental condition. Cells were plated on four-well dishes (Greiner Bio-One, 627170) pre-coated with poly-ornithine/laminin (Sigma-Aldrich, L2020-1MG).

Cells were cultured in the presence of the neurotrophic factor BDNF (10 ng/ml), and the medium was changed every second day. Precipitation experiments were performed on day 7. 16 h before precipitation, the medium was replaced by horse serum and BDNF-depleted medium (0.5 ml/well).

## NSC34 cell culture
NSC34 cells were cultured in Dulbecco's modified Eagle's medium high glucose Glutamax (Gibco, 61965059) containing 10% FCS and 100 µg/mL Penicillin/Streptomycin. For lentiviral transduction, NSC34 cells were incubated with viral particles for 15 min at RT directly before seeding and were transduced a second time after one week. For precipitation experiments and pharmacological assays, 600.000 cells were seeded on six-well dishes pre-coated with collagen (Gibco, A1048301). The next day, the medium was replaced by FCS-depleted medium 16 h before precipitation or pharmacological assay (1.5 ml/well).

## Human neuroprogenitor cell (NPC) culture
NPCs were cultured as previously described[60]. Briefly, NPC cultures were split about once every week with Accutase (Thermo Fisher,

#07920) and expanded for at least 15 passages to achieve pure NPC cultures. NPCs were cultured in neuronal medium consisting of Neurobasal medium (Gibco, #21103049), Dulbecco's modified Eagle's medium F-12 (DMEM/F-12) (Gibco, #21331046), MACS NeuroBrew-21 (Miltenyi Biotech, #130-097-263), N-2 Supplement (Gibco, #17502048) and 100 μg/mL Penicillin/Streptomycin/Glutamax (Gibco, #10378016) supplemented with 3 μM CHIR99021, 0.5 μM PMA, and 150 μM Ascorbic acid (AA) (Sigma, # A92902) was changed every other day.

## Human MN culture

For NPC differentiation into MNs, cells were cultured for 9 days in neuronal medium supplemented with 1 μM PMA. On day 2, 1 μM Retinoic acid (Stemcell Technologies, # 72264) was added to the medium. The medium was changed every other day. After 9 days, NPCs were seeded on six-well dishes pre-coated with Matrigel and the medium was switched to neuronal medium supplemented with 10 ng/mL glia-derived neurotrophic factor (GDNF) (Alomone Labs, # G-240), 10 ng/mL brain-derived neurotrophic factor (BDNF) (PeproTech, #450-02), and 500 μM dibutyryl-cAMP (dbcAMP) (Stemcell Technologies, #73886). Experiments were performed on day 14. 16 h before precipitation, the medium was replaced by MACS NeuroBrew-21 and small molecule depleted-neuronal medium (1.5 ml/well). Control iPSC-line #1 was provided by Prof. Chandran. MNs derived from the iPSC-line IMR90-4 (control iPSC-line #2), originally purchased from WiCell[78], were provided by the lab of Prof. Seeger as pre-differentiated cryostocks. Both iPSC-lines with ALS-linked SOD1 mutations (SOD1[R115G] and SOD1[D90A]) and the corresponding SOD1[D90A] isogenic control were provided by Prof. Herrmann as NPCs. The iPSC lines used in this study are listed in Supplementary Table 1.

## Trichloroacetic acid (TCA) precipitation

The secretion assay was adapted from Cruz-Garcia et al. with minor modifications[14]. Briefly, 1.5 ml or 0.5 ml cell culture medium was collected. Cells were washed with PBS, lysed in 2× Laemmli and boiled at 99 °C for 5 min. The collected medium was centrifuged first at $1000 \times g$ for 5 min and subsequently at $10,000 \times g$ for 30 min. 30 μg of BSA (Thermo Scientific) carrier protein and 0.2 ml of 100% trichloroacetic acid (TCA) were added to 1.5 ml medium and were mixed properly. For precipitation experiments with mouse MN medium no BSA was added. Proteins were precipitated on ice at 4 °C for 1 h and centrifuged $16,000 \times g$ for 30 min. The pellet was resuspended in 30 μl 2× Laemmli and boiled at 99 °C for 5 min.

## Pharmacological assay

After 16 h withdrawal of FCS, the medium was replaced by 1.5 ml fresh supplement-depleted medium with pharmacological inhibitors for 8 h. The following pharmacological inhibitors were used: MG-123 (10 μM, Enzo Life Science, BML-PI102-0005), 3-Methyladenine (5 μM, MedChem Express, HY-19312), Aloxistatin (E64d) (10 μg/ml, MedChem Express, HY-100229), Pepstatin A Methyl Ester (10 μg/ml, Merck, 516485), GW4869 (10 μM, Sigma-Aldrich, D1692). For HBSS (Gibco, 14170138) treatment, the medium was replaced by HBSS for 1 h. Proteins were precipitated as described above. The pellet was resuspended in 30 μl 2× Laemmli and boiled at 99 °C for 5 min.

## Western blot

Protein lysates were separated by SDS-PAGE and transferred to PVDF Membranes (120 V, 45 min, 4 °C). Membranes were blocked in TBS-T with 5% milk powder for 2 h at RT, probed with primary antibodies overnight at 4 °C, and incubated with horseradish peroxidase-conjugated secondary antibodies for 1 h at RT. Membranes were washed three times in TBS-T for 15 min and incubated for 5 min with developer reagents (Immobilon, WBKLS0500). Primary antibodies used for Western blot analysis are listed in Supplementary Table 5. Peroxidase-conjugated secondary antibodies against mouse, rabbit,

chicken, rat and guinea pig were obtained from Jackson Immuno-Research Laboratories. Peroxidase-conjugated secondary antibody against goat was obtained from Merck. Uncropped and unprocessed scans of all Western blots are provided in the Source Data file.

## Quantification of Sod1 secretion by Western blot

To examine the Sod1 secretion by Western blot, we loaded 20-fold the material of the medium sample compared to the lysate sample. The quantification of Sod1 secretion was performed as previously described by calculating the ratio between the amount of Sod1 in the medium and in the lysate[14]. Subsequently, the ratios were normalized to the control and set to 1 in each experiment.

## Lamp1 cell surface staining

The protocol was adapted and modified from Andrews[79]. Primary MNs were deprived of horse serum for 16 h before treatment with ionomycin. Briefly, a metal plate was cooled down on ice 20 min before the start of the experiment. Cells were washed thrice with 37 °C neurobasal medium (NBS). During the last wash cells were treated for 5 min with 37 °C NBS containing 1 mM $CaCl_2$ and 10 μM ionomycin to trigger intracellular $Ca^{2+}$ elevation and lysosomal exocytosis. Following this treatment, the cells were immediately treated with ice-cold NBS. Subsequently, the dishes were placed on the cold metal plate to inhibit endocytosis, keeping the luminal epitopes of Lamp1 on the cell surface. Incubation with the anti-Lamp1 mAb solution was carried on the metal plate for 1 h at 4 °C. The dishes were washed with ice-cold PBS three times and fixed with 4% PFA. The dishes were removed from the cold metal plate and left for 1 h at room temperature. Further ICC staining was performed according to the protocol mentioned above.

## Subcellular membrane fractionation

The protocol was adapted and modified from Zhang et al.[42]. NSC34 cells ($10 \times 10^6$ cells/dish in four 15-cm dishes) were cultured to confluence and starved in HBSS for 1 h. Cells were harvested by scraping, and homogenized with 5 ml B1 lysis buffer (20 mM HEPES-KOH, pH 7.2, 400 mM sucrose, 1 mM EDTA, 0.3 mM DTT, and a combination of cOmplete™ Protease Inhibitor Cocktail and PhosSTOP™) (Roche) through a 22 G needle (5 ml syringe) until cell lysis of ~90% was achieved, as visualized by Trypan blue staining. The homogenates were subjected to differential centrifugation at $3000 \times g$ (20 min), $25,000 \times g$ (30 min), and $100,000 \times g$ (45 min, NVT 65 rotor) to collect subcellular fractions at each step. The 25k pellet was further separated by sucrose step gradient centrifugation by resuspending in 1.28 ml of 1.25 M sucrose buffer and overlaying it with 0.85 ml of 1.1 M and 0.5 ml of 0.85 M sucrose buffer. Centrifugation was performed at $120,000 \times g$ for 2 h at 4 °C (SW32Ti rotor). The membrane layer at the boundary between 0.25/1.1 M (L fraction) and pellet (P) was collected. The L fraction was further resuspended in 19% Optiprep solution, and step gradients containing the following layers were generated-0.5 ml 22.5%, 1 ml 19% (sample), 0.9 ml 16%, 0.9 ml 12%, 1 ml 8%, 0.5 ml 5% and 0.2 ml 0%. Each density of OptiPrep was prepared by diluting 50% OptiPrep (20 mM Tricine-KOH, pH 7.4, 42 mM sucrose and 1 mM EDTA) with a buffer containing 20 mM Tricine-KOH, pH 7.4, 250 mM sucrose and 1 mM EDTA. The OptiPrep gradient was centrifuged at $150,000 \times g$ for 3 h (SW32Ti rotor), and subsequently, nine fractions were collected from the top. Fractions were diluted with B88 buffer (20 mM HEPES-KOH, pH 7.2, 250 mM sorbitol, 150 mM potassium acetate, and 5 mM magnesium acetate) and membranes were collected by centrifugation at $100,000 \times g$ for 1 h. All membrane fractions were normalized to their membrane phosphatidylcholine content and analyzed by western blots.

## Immunoisolation of 3xFLAG-LC3

NSC34 cells ($10 \times 10^6$ cells/dish, five 15-cm dishes) transduced with FUV-3xFLAG-LC3B were maintained at confluence for 2 days and starved with HBSS for 1 h. The $25,000 \times g$ membrane pellet was

collected as described above. The pellet was resuspended in 1 ml of immunoisolation buffer (25 mM HEPES, pH 7.4, 140 mM potassium chloride, 5 mM sodium chloride, 2.5 mM magnesium acetate, 50 mM sucrose, and 2 mM EGTA). Resuspended pellets were divided into two equal portions containing Flag M2 beads with or without 0.2 mg/ml 3xFLAG tag-blocking peptides and mixed overnight at 4 °C in an end-to-end rotor. Membranes associated with the beads were washed three times with immunoisolation buffer and eluted twice by the addition of 0.5 mg/ml 3xFLAG peptide for 0.5 h at RT. Eluates were centrifuged at $100,000 \times g$ for 50 min at 4 °C and analyzed by Western blot.

## Preparation of detergent-soluble and -insoluble fractions
Mouse spinal cord were dissected and frozen in liquid nitrogen. Tissues were homogenized in five volumes of ice-cold 0.25 M sucrose buffer (50 mM Tris-HCl pH 7.4, 1 mM EDTA) supplemented with a cOmplete™ Protease Inhibitor Cocktail (Roche). Homogenates were centrifuged at $500 \times g$ for 10 min at 4 °C. Subsequently, supernatants were collected and lysed with an equal volume of cold sucrose buffer containing 1% Triton X-100. Lysates were subjected to centrifugation at $13,000 \times g$ for 15 min at 4 °C to separate supernatants (Triton-X-100 soluble fraction) and pellets. Pellets were resuspended in 1% SDS in PBS (Triton-X-100-insoluble fractions).

## Preparation of tissue lysates for lysosomal enzyme activity assays
Spinal cord tissue was homogenized in 20 volumes (vol) of 50 mM Tris HCl 50 mM NaCl and 0.1% Triton X100 (vol/vol), incubated on ice for 30 min, and cleared by centrifugation at 13000 rpm for 20 min at 4 °C. The specific activity of lysosomal enzymes (β-hexosaminidase, β-glucuronidase, and β-galactosidase) was determined colorimetrically with monosaccharide substrates coupled to p-nitrophenyl under acidic conditions. For β-glucuronidase and β-galactosidase, 50 µl of the lysate was incubated with the colorimetric substrates (p-nitrophenyl-β-D-glucuronide and p-nitrophenyl-2-acetamido-2-deoxy-beta-D-glucopyranoside, respectively) for 4 h in a 0.1 M potassium-citrate buffer at pH 4.6, containing 0.08% NaN3, and 0.2% BSA. For β-hexosaminidase activity determination, 10 µl of the lysate was incubated for 30 min with the substrate (p-nitrophenyl-2-acetamido-2-deoxy-β-D-glucopyranosid) in the same buffer. The reaction was stopped by the addition of 0.4 M Glycin/NaOH pH 10.4, and the absorption was measured at a wavelength of OD405 nm in a 96-well plate reader.

## Plasmid construction
To generate the plasmids for the knockdown of Atg9, Stx17, Snap29, Snap23, PLEKHG5 two short hairpin (sh)-sequences were synthesized as sense and antisense oligos and cloned into the BamHI and EcoRI restriction sites of pSIH-H1-eGFP. The individual oligo sequences are listed in Supplementary Table 6.

To generate FUV-3xFLAG-LC3 we cloned the 3xFLAG-LC3 cassette of CMV-3xFLAG-LC3B[80] into the backbone of the FUV-HA-TDP43-WT[81]. CMV-3xFLAG-LC3 was digested by KpnI, and SmaI, and the resulting 1.3 kB insert was purified using the NucleoSpin Gel and PCR Clean-up kit (Macherey Nagel) and blunted using the Anza™ DNA End Repair kit (Thermo Scientific). FUV-HA-TDP43 was digested by BamHI, and EcoRI, and the 10 kB backbone was purified using NucleoSpin Gel and PCR Clean-up kit (Macherey Nagel), blunted using the Anza™ DNA Blunt End Kit (Thermo Scientific) and dephosphorylated with FastAP (Thermo Scientific). Subsequently, both fragments were ligated with T4 DNA Ligase (Thermo Scientific).

## Lentivirus production and transduction
Lentiviral particles were produced by co-transfecting HEK293T cells with the indicated expression and packaging plasmids by calcium phosphate precipitation. The medium was replaced 16 h after transfection. 48 h after transfection, the medium was collected, and the virus was concentrated by ultracentrifugation. For lentiviral transduction, the cells were incubated with viral particles for 10 min at RT directly before plating. Plasmids used for lentivirus production are given in Supplementary Table 3.

## RNA extraction
Dissected spinal cord tissues were frozen in liquid nitrogen and RNA was extracted using the RNeasy Lipid Tissue Mini Kit (Qiagen) according to the manufacturer's instructions. To remove genomic DNA, we performed on-column DNase digestion with the RNase-Free DNase Set (Qiagen) according to manufacturer's instructions.

## cDNA synthesis & qPCR
1 µg RNA was reverse transcribed with random hexamers using the first Strand cDNA Synthesis Kit (Thermo Scientific) according to the manufacturer's instructions. Reverse transcription reactions were diluted 1:5 in water and qPCR reactions were set up with the Luminaris HiGreen qPCR Master Mix (Thermo Fisher Scientific) on a LightCycler® 96 (Roche). Relative expression was calculated using the ΔΔCt method. Primer sequences are given in Supplementary Table 6.

## RNAseq
RNA quality was checked using a 2100 Bioanalyzer with the RNA 6000 Nano kit (Agilent Technologies). The RIN for all samples was ≥8.7. DNA libraries suitable for sequencing were prepared from 500 ng of total RNA with oligo-dT capture beads for poly-A-mRNA enrichment using the TruSeq Stranded mRNA Library Preparation Kit (Illumina) according to manufacturer's instructions. After 12 cycles of PCR amplification, the size distribution of the barcoded DNA libraries was estimated ~310 bp by electrophoresis on Agilent DNA 1000 Bioanalyzer microfluidic chips.

Sequencing of pooled libraries, spiked with 1% PhiX control library, was performed at 33 million reads/sample in single-end mode with 75 nt read length on the NextSeq 500 platform (Illumina) using a High Output sequencing kit. Demultiplexed FASTQ files were generated with bcl2fastq2 v2.20.0.422 (Illumina).

To assure high sequence quality, Illumina reads were quality- and adapter-trimmed via Cutadapt[82] version 2.4 using a cutoff Phred score of 20 in NextSeq mode, and reads without any remaining bases were discarded (command line parameters: --nextseq-trim=20 -m 1 -a AGATCGGAAGAGCACACGTCTGAACTCCAGTCAC). Processed reads were subsequently mapped to the mouse genome (GRCm38.p6) and the cDNA sequence of human SOD1 (NM_000454.4) using STAR[83] v2.7.2b with default parameters based on mouse RefSeq annotation version 108 combined with SOD1 annotation. Read counts on the exon level summarized for each gene were generated using featureCounts v1.6.4 from the Subread package[84]. Multi-mapping and multi-overlapping reads were counted strand-specific and reversely stranded with a fractional count for each alignment and overlapping feature (command line parameters: -s 2 -t exon -M -O --fraction). The count output was utilized to identify differentially expressed genes using DESeq2[85] version 1.24.0. Read counts were normalized by DESeq2, and fold-change shrinkage was applied by setting the parameter "betaPrior=TRUE". Differential expression of genes was assumed at an adjusted $p$-value (padj) after Benjamini–Hochberg correction <0.05.

## Statistics and reproducibility
All statistical analyses were performed using GraphPad Prism version 9 for Windows (GraphPad Software, San Diego, California USA) and statistical significance was considered at test level $P < 0.05$. Quantitative data are presented as mean ± SEM unless otherwise indicated. No statistical method was used to predetermine sample size. No data were excluded from the analyses. Chosen statistical tests are individually listed in corresponding figure legends. Most experiments were

carried out independently at least three times and individual data points are presented in all graphs with $n < 10$. Details of replicate numbers, quantification and statistics for each experiment are specified in the figure legends. The experiments were not randomized and investigators were not blinded to allocation during experiments and outcome assessment.

## Reporting summary

Further information on research design is available in the Nature Portfolio Reporting Summary linked to this article.

## Data availability

Source data are provided with this paper. The RNA-seq data have been deposited in NCBI's Gene Expression Omnibus (GEO) with dataset identifier GSE269588. Source data are provided with this paper.

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

## Acknowledgements

We thank Janina Dix for excellent technical support. We are grateful to Hilde Troll for lentiviral and AAV vector production. We thank Regine Sendtner and Sebastian Rose for monitoring of the transgenic SOD1 animals. P.L. was supported by the BMBF grants VORAN, 161L0150 and VORAN-2, 16LW066, and the DFG grant DFG LU 2347/3-1. T.B. was supported by the IZKF Würzburg project Z-6. The JEOL JEM-2100 transmission electron microscope is funded by the Deutsche Forschungsgemeinschaft (DFG, German Research Foundation) – 218894163 and the structured illumination microscope Zeiss Elyra S.1 SIM is funded by DFG – 261184502 (INST 93/823-1 FUGG).

## Author contributions

P.L. conceived and supervised the study. A.J.H., B.H. and A.V. performed most of the experiments, data analysis and statistical analysis. N.J.G., A.Z., C.A., T.O., M.Sch., M.D. contributed to several experiments and data analyses. M.B., T.B. bioinformatic analysis. A.H, B.S., B.T.S., S.C.,

J.S., S.P., provided hiPSC lines. C.S. supervised electron microscopy. P.L. wrote the manuscript. M.S. reviewed and revised the manuscript. P.L. and M.S. provided financial support. All authors approved the final version of the manuscript.

## Funding

## Competing interests
The authors declare no competing interests.

## Additional information

[1]Institute of Clinical Neurobiology, University Hospital Würzburg, Würzburg, Germany. [2]Institute for Food Quality and Safety, Research Group Food Toxicology and Alternative/Complementary Methods to Animal Experiments, University of Veterinary Medicine Hannover, Hannover, Germany. [3]Bloomberg School of Public Health, Center for Alternatives to Animal Testing, Johns Hopkins University, Baltimore, MD, USA. [4]Centre for Clinical Brain Sciences, University of Edinburgh, Edinburgh EH16 4SB, UK. [5]UK Dementia Research Institute at University of Edinburgh, University of Edinburgh, Edinburgh EH16 4SB, UK. [6]Anne Rowling Regenerative Neurology Clinic, University of Edinburgh, Edinburgh EH16 4SB, UK. [7]Center for Regenerative Therapies TU Dresden, Fetscherstr. 105, 01307 Dresden, Germany. [8]Medical Faculty Carl Gustav Carus of TU Dresden, Dresden, Germany. [9]Department of Neurology, Hannover Medical School, Hannover, Germany. [10]Imaging Core Facility, Biocenter, University of Würzburg, 97074 Würzburg, Germany. [11]Core Unit Systems Medicine, University of Würzburg, D-97080 Würzburg, Germany. [12]Translational Neurodegeneration Section Albrecht-Kossel, Department of Neurology, University Medical Center Rostock, Rostock, Germany. [13]Center for Transdisciplinary Neurosciences Rostock, University Medical Center Rostock, Rostock, Germany. [14]Deutsches Zentrum für Neurodegenerative Erkrankungen (DZNE) Rostock/Greifswald, 18147 Rostock, Germany. [15]Institute of Biochemistry, Christian-Albrechts-University Kiel, Olshausenstr. 40, 24098 Kiel, Germany. [16]These authors contributed equally: Amy-Jayne Hutchings, Bita Hambrecht, Alexander Veh. ✉e-mail: Lueningsch_P@ukw.de

