## [Peer Review File · Nature Communications]

Plekhg5 controls the unconventional secretion of Sod1 by presynaptic secretory autophagyREVIEWER COMMENTS

Reviewer #1 (Remarks to the Author):

In this study, the authors have uncovered a significant role for the rab26 GEF Plekhg5 in the unconventional secretion of Sod1. Utilizing cultured motoneurons, cell lines, and Plekhg5 mutant mice, they have convincingly demonstrated the importance of Plekhg5 in this process. Furthermore, their findings indicate that deletion of Plekhg5 extends the survival of mutant SOD1G93A mice, attributable to reduced microglial activation stemming from decreased Sod1 secretion. Notably, this mechanism likely extends to other neurodegenerative diseases, challenging the notion of strict cell autonomy, a notion that the authors could discuss further.

The results presented in this article are both novel and highly significant within the realms of unconventional secretion and neurodegeneration. The methodology employed was robust, encompassing rigorous acquisition and quantification of data. Furthermore, the presentation and discussion of the findings are clear and comprehensive.

However, to further support the claims of the authors in this manuscript, it would be important to include additional genetic experiments targeting the SNAREs implicated in autophagosome-lysosome fusion (Stx17, Snap29, Vamp7/Vamp8) and LRO secretion (Vamp7, Snap23). Such experiments would enhance the mechanistic understanding and broaden the scope of the study's implications.

Reviewer #2 (Remarks to the Author):

In their manuscript, Hutchings et al. elucidate a pathophysiological mechanism involving two proteins associated with motoneuron diseases, Plekhg5 and Sod1. They demonstrate that Plekhg5 activates the small GTPase Rab26, leading to the exocytosis of Sod1 sequestered in lysosomal-related organelles (LROs). Additionally, the depletion of Plekhg5 in the ALS model of SODG39A mice accelerates disease onset but decelerates disease progression. These findings are further corroborated in ALS-patient derived motor neurons.

The identification of a pathway on which two motoneuron disease-associated proteins converge makes the findings certainly interesting for the field. However, additional experiments and clarifications should be considered prior to publication:

General:

The authors primarily employ Western Blot analysis to quantify changes in protein levels and base their conclusions on these results. However, the absence of protein loading controls such as actin or tubulin for media samples compromises the reliability of their findings. To enhance the credibility of the presented results, the authors could either include total protein visualization to ensure equal loading for each lane or utilize alternative experimental approaches to validate their conclusions. Moreover, while the authors claim to have loaded the same amount of total protein in each lane, discrepancies in detected actin or tubulin bands across several images suggest variable protein loading between SDS-PAGE lanes (e.g., Figure 2A, 2C, 3G, 7A, 7B, 7D). Protein quantification should be adjusted for differences in total protein loading before normalizing protein levels to control samples. It remains unclear from the methods description whether the authors accounted for these differences.

Figure 2:

In Figure 2F, the authors utilize compartmentalized chambers to assess intra- and extracellular Sod1 levels in the somatodendritic versus axonal compartment. However, a quantification is missing here. While assuming these blots are representative and that a comparable amount of total protein was loaded, the image effectively demonstrates that Plekhg5 depletion abolishes Sod1 secretion from the axonal compartment, as described in the text. Nonetheless, this image raises several concerns. Firstly, somatic Sod1 secretion appears significantly higher than axonal Sod1 secretion. Furthermore, somatic Sod1 secretion remains unchanged in Plekhg5-deficient cells. Given these observations, it is questionable whether the >50% reduction in Sod1 protein levels in the medium of Plekhg5-deficient cells presented in Figures 2D and 2E can be attributed to the low amounts of Sod1 secreted from axonal compartments under control conditions. Additionally, the reduced microglia activation in the Plekhg5-deficient ALS mouse model is surprising if, as suggested by this image, Sod1 is secreted from the somatodendritic compartment at relatively high levels and independently of Plekhg5. Moreover, Sod1 levels are intracellularly elevated in both compartments, contrary to the axon-specific Sod1 protein accumulation described in Figure 4. In Figures 2G-L, the authors demonstrate that Plekhg5 mediates Sod1 secretion through Rab26 activation using Rab26 shRNAs and a continuous active form of Rab26. They should consider including controls demonstrating unaltered Sod1 protein expression upon Rab26 knockdown/activation.

Figure 7:

Figure 7 illustrates PLEKHG5-driven secretion of mutant SOD1 in ALS patient-derived motor neurons. Neuronal differentiation protocols guided by small molecules often exhibit significant variability between batches. To ensure consistent conditions across different lines and experiments, differentiation efficiency and expression of motor neuron markers should be assessed. The authors should provide additional information on quality control measures

implemented to ensure consistent conditions among the neuronal cultures used for the results presented in Figure 7.

In Plekhg5-deficient mice, immunohistochemistry results indicate Sod1 accumulation at presynaptic sites. Is this observation also seen in human neurons? Furthermore, the results suggest an internal feedback mechanism that downregulates PLEKHG5 levels to prevent excessive secretion of mutant SOD1. This raises the question whether a similar downregulation of Plekhg5 levels is observed in Sod1 mice?

Reviewer #3 (Remarks to the Author):

In this study, the authors connect Plekhg5 function to the accumulation and secretion of Sod1, a protein involved in ALS. Previous studies showed that Plekhg5, linked to several other forms of motoneuron disease, regulates autophagy of synaptic vesicles via activation of Rab26. In this study, the authors show that Plekhg5 regulates Sod1 secretion through a mechanism that involves an autophagosome intermediate and a LAMP1+ lysosome related organelle. Overall, the study makes an important new disease-related contribution, is well executed and supported by data in several cell line and animal models. As detailed below, some aspects of the study require further explanation and/or quantitation.

1. Figure S1 is missing staining in control Plekhg5+/+ mice. The authors conclusions are not obvious (based on the current images) and require the control for comparison.
2. Figure 1I and 1J: It is not clear how the Sod1/Tuj1 ratios were calculated for the SDS soluble blot. Based on the blot image, there are considerably greater levels of Sod1 compared to Tuj1 (barely detectable), but this is not represented in the corresponding bar graph.
3. Figure 2H and 2I: The sh-Rab26#1 and #2 labels don't appear to correspond to the data shown in the corresponding western blot (ie. #2 shows a greater knockdown effect by western but that is not reflected in the bar graph). Are colours reversed?
4. Figure 2L is missing a negative control for the rescue effects and for Rab26 activation (eg. Empty vector and wild-type or inactive Rab26). One of the main conclusions is that "This secretory pathway depends on the activation of the small GTPase Rab26 by Plekhg5", so this statement requires better support.

5. Figure 3 nicely shows dependence on Atg5, and somewhat on Atg9, for secretion of Sod1 into the medium. To determine whether this mechanism also includes distal components of secretory autophagy, it would be useful to similarly test STX3, STX4 or SNAP29.

6. P. 7, last paragraph would benefit from clarification. “In this scenario, Sod1 would be released as a free protein”. Since the previous sentence is referring to fusion of the outer autophagosomal membrane with a lysosome, does this mean “released into the autolysosome”? Rather, I think the authors intend to indicate released into the media as a free protein versus released into the media within a sEV.

7. Figure 5B: BafA1 experiment: if BafA1 blocks the fusion of autophagosomes with lysosomes (or LROs), then what is the proposed mechanism (ie. secretory route) for the enhanced Sod1 secretion upon treatment with BafA1? This finding appears inconsistent with the authors proposed model.

8. The rationale and conclusions regarding Figure 5E-F are unclear. The authors conclude that these images show that p62 and ubiquitin were absent from Lamp1+ vesicle clusters, “confirming that the global proteostasis is unaffected by the depletion of Plekhg5”. From the images, it appears that p62 and Lamp1+ co-localize in some instances. The description and/or images require clarification and also quantitation.

9. Figure 6L requires quantitation, especially to substantiate the difference between Plekhg5^{+/+} SOD1[G93A] and Plekhg5^{-/-} SOD1[G93A].

Other:

p.31. Figure 2 legend is missing a title.

Figure 6D: y axis label should be survival, not onset.

We thank all reviewer for their constructive feedback. Below, please find the point-by-point answers to the individual comments. **The answers are highlighted by red font.**

Reviewer #1 (Remarks to the Author):

In this study, the authors have uncovered a significant role for the rab26 GEF Plekhg5 in the unconventional secretion of Sod1. Utilizing cultured motoneurons, cell lines, and Plekhg5 mutant mice, they have convincingly demonstrated the importance of Plekhg5 in this process. Furthermore, their findings indicate that deletion of Plekhg5 extends the survival of mutant SOD1G93A mice, attributable to reduced microglial activation stemming from decreased Sod1 secretion. Notably, this mechanism likely extends to other neurodegenerative diseases, challenging the notion of strict cell autonomy, a notion that the authors could discuss further.

We followed the reviewer's suggestion and extended the discussion on the non-cell autonomy and the involvement of autophagy in this regard. The following passage was incorporated into the discussion: "Our findings add further weight to the notion that motoneuron degeneration in ALS does not only involve cell-intrinsic mechanisms but also depends on non-cell autonomous mechanisms by other cell types [1]. In addition, our data support the idea that unconventional pathways for protein secretion, such as secretory autophagy, are critically involved in the pathophysiology of neurodegenerative diseases, which contribute to non-cell autonomous mechanisms of neurodegeneration. Recent studies showed that lysosomal exocytosis releases pathogenic alpha-Synuclein species, that the secretion of wildtype and mutant Tau depends on secretory autophagy, and that the mutant Huntington is secreted via unconventional secretory pathways [2-4]. Secretion of such pathogenic proteins triggers microglial neuroinflammation, driving the non-cell autonomous neurodegeneration. Furthermore, pharmacological inhibition of PIKFYVE kinase activates a UPS pathway involving exocytosis of aggregation-prone proteins. Furthermore, inhibition of PIKFYVE ammoniated the phenotype of several ALS models [5]."

The results presented in this article are both novel and highly significant within the realms of unconventional secretion and neurodegeneration. The methodology employed was robust, encompassing rigorous acquisition and quantification of data. Furthermore, the presentation and discussion of the findings are clear and comprehensive.

We thank the reviewer for his positive feedback.

However, to further support the claims of the authors in this manuscript, it would be important to include additional genetic experiments targeting the SNAREs implicated in autophagosome-lysosome fusion (Stx17, Snap29, Vamp7/Vamp8) and LRO secretion (Vamp7, Snap23). Such experiments would enhance the mechanistic understanding and broaden the scope of the study's implications.

As suggested by the reviewer, we genetically targeted Stx17 and Snap29 to block autophagosome-lysosome fusion and Snap23 to block LRO secretion. Targeting of Stx17, Snap29, and Snap23 by sh-RNA resulted in reduced levels of Sod1 released into the cell culture medium, indicating the involvement of these SNAREs in the Plekhg5/Rab26-mediated secretion of Sod1. These data confirm that the fusion between autophagosomes and lysosomes/LROs appears as an essential step for the secretion of Sod1. We incorporated this new data set into the revised version of Fig. 4 (see below).

Figure 4

J-L: Western blots showing the knockdown of Stx17 (J), Snap29 (K), and Snap23 (L) upon simultaneous lentiviral expression of two different sh-RNAs in primary MNs.

M-O: The Sod1 secretion is blocked upon knockdown of Stx17 (M), Snap29 (N), and Snap23 (O). Western blots of lysates and media of primary MNs transduced with sh-RNAs targeting the indicated SNAREs.

P: Quantification of the Sod1 intensity in lysates and media. sh-Stx17, n=4; sh-Snap29, n=6; sh-Snap23, n=6. One-sample t-test. All data are shown as mean ± SEM.

*p % 0.05; ***p % 0.001.

Reviewer #2 (Remarks to the Author):

In their manuscript, Hutchings et al. elucidate a pathophysiological mechanism involving two proteins associated with motoneuron diseases, Plekhg5 and Sod1. They demonstrate that Plekhg5 activates the small GTPase Rab26, leading to the exocytosis of Sod1 sequestered in lysosomal-related organelles (LROs). Additionally, the depletion of Plekhg5 in the ALS model of SODG39A mice accelerates disease onset but decelerates disease progression. These findings are further corroborated in ALS-patient derived motor neurons.

The identification of a pathway on which two motoneuron disease-associated proteins converge makes the findings certainly interesting for the field. However, additional experiments and clarifications should be considered prior to publication:

General:

The authors primarily employ Western Blot analysis to quantify changes in protein levels and base their conclusions on these results. However, the absence of protein loading controls such as actin or tubulin for media samples compromises the reliability of their findings. To enhance the credibility of the presented results, the authors could either include total protein visualization to ensure equal loading for each lane or utilize alternative experimental approaches to validate their conclusions. Moreover, while the authors claim to have loaded the same amount of total protein in each lane, discrepancies in detected actin or tubulin bands across several images suggest variable protein loading between SDS-PAGE lanes (e.g., Figure 2A, 2C, 3G, 7A, 7B, 7D). Protein quantification should be adjusted for differences in total protein loading before normalizing protein levels to control samples. It remains unclear from the methods description whether the authors accounted for these differences.

We apologize for not providing detailed information on how we quantified the Sod1 levels in the medium and lysate of the individual samples. Indeed, the information we provided in the original

manuscript was not sufficient. We would like to emphasize that we did not only normalize the intensities of Sod1 to the control. Quantification of the Sod1 secretion was calculated as the ratio between Sod1 in the medium and Sod1 in the lysate from the same cell culture. After determining this ratio, the Sod1 levels were normalized to the control. The intensity of Sod1 in lysates was adjusted to Tuj1 (primary MNs) or Actin (NSC34 cells), before normalizing to the control.

We followed previously described procedures to determine the Sod1 levels in the medium, which have been applied by the Malhorta lab to analyze Sod1 [6], but also by the Schekman lab to analyze the secretion of IL-1 β [7] or FABP4 [8]. Calculation of the ratio between medium and lysate gives a more accurate measurement of how much of the total Sod1 is secreted, in contrast to analyzing the total Sod1 amount in the medium. Most likely many unconventionally secreted proteins are affected by this pathway making it difficult to determine an appropriate loading control or normalizing to the total protein in the medium.

In the revised version of the manuscript, we included a detailed description of the procedure in the corresponding figure legends and the methods section.

Figure 2:

In Figure 2F, the authors utilize compartmentalized chambers to assess intra- and extracellular Sod1 levels in the somatodendritic versus axonal compartment. However, a quantification is missing here. While assuming these blots are representative and that a comparable amount of total protein was loaded, the image effectively demonstrates that Plekhg5 depletion abolishes Sod1 secretion from the axonal compartment, as described in the text. Nonetheless, this image raises several concerns. Firstly, somatic Sod1 secretion appears significantly higher than axonal Sod1 secretion. Furthermore, somatic Sod1 secretion remains unchanged in Plekhg5-deficient cells. Given these observations, it is questionable whether the >50% reduction in Sod1 protein levels in the medium of Plekhg5-deficient cells presented in Figures 2D and 2E can be attributed to the low amounts of Sod1 secreted from axonal compartments under control conditions. Additionally, the reduced microglia activation in the Plekhg5-deficient ALS mouse model is surprising if, as suggested by this image, Sod1 is secreted from the somatodendritic compartment at relatively high levels and independently of Plekhg5. Moreover, Sod1 levels are intracellularly elevated in both compartments, contrary to the axon-specific Sod1 protein accumulation described in Figure 4.

We thank the reviewer for pointing out this important aspect. To address this point, we performed additional experiments, included quantifications, and revised the corresponding results section. As correctly pointed out by the reviewer, the initial presentation of the data suggested a stronger secretion of Sod1 in the somatodendritic compartment. However, this has technical reasons. To obtain comparable intensities of Sod1 in the lysate and medium within the same exposure time on the same blot, we loaded 20-fold the material of the cell culture medium sample compared to the lysate sample (a comparable ratio has previously been described [6]). However, the medium samples from the axonal side were loaded only with a 4-fold enrichment in comparison to the lysate from the same compartment. Therefore, it was difficult to conclude from these data whether there is a higher secretion rate on the axonal side. We apologize for not including this important information.

In the new experiments, we loaded the medium samples from both sides of the microfluidic chambers with the same 20-fold enrichment of the medium samples for better comparison (Fig. 2 F). To quantify the secretion of Sod1, we calculated the ratio between the Sod1 intensity in the medium and lysate

and normalized this ratio to the control (sh-Luc) of the somatodendritic side (Fig. G, H). This quantification revealed a clear enrichment of Sod1 in the medium of the axonal side, suggesting a more efficient secretion in axons. Upon depletion of Plekhg5, we detected a significant reduction of Sod1 in the medium of the axonal side (Fig. 2 H). On the somatodendritic side, we also observed a decrease of Sod1 in the medium upon knockdown of Plekhg5. However, this difference did not reach statistical significance (Fig. 2 H).

To quantify the Sod1 levels in the lysate, the Sod1 intensity was adjusted to Tuj1 and normalized to the control (sh-Luc) lysate from the somatodendritic side. In contrast to the enrichment of Sod1 in the medium of the axonal side, Sod1 was enriched in the lysate of the somatodendritic side. Knockdown of Plekhg5 caused a significant increase of Sod1 in the lysate of the somatodendritic side. On the axonal side, we also observed an increase, which did not reach statistical significance (Fig. 2 G).

While the axonal side of the microfluid chambers represents pure axons, the somatodendritic side also contains axons, at least to a certain extent. These axons will also contribute to the effects observed on the somatodendritic side of the chamber, limiting the interpretation of the results obtained from this side in terms of specificity. Taking this into consideration we would like to conclude that Sod1 is enriched in the medium of the axonal side and that Plekhg5 depletion blocks the secretion of Sod1 in axons. We revised the figure and the corresponding results section accordingly.

Figure 2

F: Western blot showing the Sod1 levels in lysates and media of the somatodendritic and axonal compartment. HE, high exposure; LE, low exposure.

G, H: Western blot quantifications of the Sod1 levels in lysate (G) and medium (H) upon knockdown of Plekhg5 in primary MNs cultures in compartmentalized microfluid chambers. n=4. Two-Way ANOVA, Šídák's multiple comparisons test.

Quantification of Sod1 secretion was calculated as the ratio between the amount of Sod1 in the medium and the amount of Sod1 in the lysate. The Sod1 levels in the lysates were adjusted to the Tuj1 (MNs). Subsequently, the Sod1 levels were normalized to the control and set to 1 in each experiment.

All data are shown as mean \pm SEM. *p % 0.05; ***p % 0.001.

In Figures 2G-L, the authors demonstrate that Plekhg5 mediates Sod1 secretion through Rab26 activation using Rab26 shRNAs and a continuous active form of Rab26. They should consider including controls demonstrating unaltered Sod1 protein expression upon Rab26 knockdown/activation.

The protein expression of Sod1 upon knockdown and activation of Sod1 is shown by Western blot in Fig. 2 J (knockdown of Rab26) and Fig. 2 M (expression of Flag-Rab26-WT and Flag-Ran26-QL). The corresponding quantifications are shown in Fig. 2 K and Fig. 2 N. We detected no significant changes in the Sod1 protein expression upon Rab26 knockdown/activation.

Figure 7:

Figure 7 illustrates PLEKHG5-driven secretion of mutant SOD1 in ALS patient-derived motor neurons. Neuronal differentiation protocols guided by small molecules often exhibit significant variability between batches. To ensure consistent conditions across different lines and experiments, differentiation efficiency and expression of motor neuron markers should be assessed. The authors should provide additional information on quality control measures implemented to ensure consistent conditions among the neuronal cultures used for the results presented in Figure 7.

We followed the reviewer's suggestion and included stainings for Islet1 and Tuj1 to assess the quality and differentiation efficiency of the neuronal cultures. In the revised version of Sup. Fig. 4, we provided low magnification images of each of the individual lines used in the study (Sup. Fig. 4 C). Furthermore, we quantified the number of Islet-1⁺ cells from three independent MN differentiation (Sup. Fig. 4 B). The data show that the majority of the cells stained positive for Tuj1 and we observed 70-90% Islet1⁺ cells in our culture with no major differences between the individual lines and among the independent differentiations. In summary, these data suggest that the differentiation efficiency across the different cell lines and experiments works robustly.

Supplementary Figure 4

(B) Quantification of the percentage of Islet1⁺ cells per nuclei. Each data point represents the percentage of at least 700 cells analyzed per cell line. Three independent experiments. n=3. Mean ± SEM.

(C) iPSC-derived MNs were stained for Islet1 and Tuj1 after two weeks of maturation. Low magnification images of the immunocytochemical stainings revealed no major differences in the differentiation efficiency between the indicated iPSC-lines.

In Plekhg5-deficient mice, immunohistochemistry results indicate Sod1 accumulation at presynaptic sites. Is this observation also seen in human neurons?

We thank the reviewers for this critical suggestion and performed immunocytochemical stainings of SOD1 to assess the SOD1 abundance in axon terminals of human neurons upon Plekhg5-depletion. Quantification of the immunofluorescent intensity revealed an accumulation of SOD1 in axon terminals upon knockdown of PLEKHG5 (Fig. 7 D, E). Correlating with the phenotype in Plekhg5-deficient mice, we also observed axon terminals with a swollen morphology in human neurons with PLEKHG5 depletion (Fig. 7 D). We incorporated these new data in the revised version of Figure 7.

Figure 6

(E) Immunoreactivity of SOD1 in axon terminals of control or PLEKHG5-depleted hiPSC-derived MNs.

(F) Quantification of the SOD1 intensity normalized to GFP. sh-Luc, n=48, sh-PLEKHG5, n=52. Two-tailed test, unpaired.

Furthermore, the results suggest an internal feedback mechanism that downregulates PLEKHG5 levels to prevent excessive secretion of mutant SOD1. This raises the question whether a similar downregulation of Plekhg5 levels is observed in Sod1 mice?

To analyze a potential downregulation of Plekhg5 in SOD^{G93A} mice, we performed Western blot analysis of spinal cord lysates (Sup. Fig. 7 D, E). We did not find any differences in the Plekhg5 expression in SOD^{G93A} mice, suggesting that the internal feedback mechanism we observed in human cells might be species-specific. Another reason for this result might be a specific downregulation of Plekhg5 in motoneurons, which are strongly enriched in iPSC-derived culture and highly “diluted” in the spinal cord. Unfortunately, the lack of an antibody which reliably detects Plekhg5 in immunocytochemical stainings limits the MN-specific analysis in spinal cord cross-sections. Thus, additional experiments are required to characterize this feedback mechanism in depth. This is out of the scope of our recent study and could be an exciting topic to follow up on.

Supplementary Figure 4

(D) Western blot showing the Plekhg5 expression in spinal cord lysates of wildtype and SOD1^{G93A} mice.

(E) Quantification of the Plekhg5 expression normalized to Calnexin. n=3. Mean ± SEM.

Reviewer #3 (Remarks to the Author):

In this study, the authors connect Plekhg5 function to the accumulation and secretion of Sod1, a protein involved in ALS. Previous studies showed that Plekhg5, linked to several other forms of motoneuron disease, regulates autophagy of synaptic vesicles via activation of Rab26. In this study, the authors show that Plekhg5 regulates Sod1 secretion through a mechanism that involves an autophagosome intermediate and a LAMP1+ lysosome related organelle. Overall, the study makes an important new disease-related contribution, is well executed and supported by data in several cell line and animal models. As detailed below, some aspects of the study require further explanation and/or quantitation.

We thank the reviewer for his positive feedback on our study.

1. Figure S1 is missing staining in control Plekhg5^{+/+} mice. The authors conclusions are not obvious (based on the current images) and require the control for comparison.

We followed the reviewer's suggestions and added images from control animals. These images support our conclusions that depletion of Plekhg5 results in the clustering of Sod1, but not TDP43, Tau, or p62.

Supplementary Figure 1

(A, B) Lumbar spinal cord cross sections from wildtype and *Plekhg5*-deficient mice stained for Sod1, Tau, Tdp43 and p62. Images of the ventral horn are shown in (A). Higher magnifications are shown in (B). Note that *Plekhg5* deficiency only caused accumulations of Sod1. Scale bar upper panel: 100 μm ; Scale bar lower panel: 50 μm .

2. Figure 1I and 1J: It is not clear how the Sod1/Tuj1 ratios were calculated for the SDS soluble blot. Based on the blot image, there are considerably greater levels of Sod1 compared to Tuj1 (barely detectable), but this is not represented in the corresponding bar graph.

We thank the reviewer for pointing out the discrepancy. Since Tuj1 was barely detectable in SDS soluble fractions (as pointed out by the reviewer), we quantified the levels of SDS soluble Sod1 by normalizing to the TX soluble Tuj1 levels. For clarity, we show each of the ratios in separate graphs in the revised version of the manuscript (Fig. 1 H, I). We also labeled the Y-axes more accurately to avoid any misunderstanding.

In addition, we also calculated the ratio between SDS soluble Sod1 and TX soluble Sod1, confirming that we did not detect a significant increase in SDS soluble Sod1 in spinal cord lysates of *Plekhg5*-deficient mice.

Figure 1

(H) Accumulation of Triton-X-100 soluble Sod1 in the spinal cord of *Plekhg5*^{-/-} mice. Spinal cord homogenates were separated into Triton-X-100-soluble and SDS-soluble fractions and analyzed by Western blot.

(I) Quantification of the spinal cord Sod1 levels in the Triton-X-100-soluble and SDS-soluble fraction of *Plekhg5*^{-/-} and *Plekhg5*^{+/+} mice. n=6; t-Test, two-tailed.

All data in Figure 1 are shown as mean \pm SEM. *p % 0.05; **p % 0.01.

3. Figure 2H and 2I: The sh-Rab26#1 and #2 labels don't appear to correspond to the data shown in the corresponding western blot (ie. #2 shows a greater knockdown effect by western but that is not reflected in the bar graph). Are colours reversed?

Indeed, the colors were reversed. In the revised version we corrected this mistake.

4. Figure 2L is missing a negative control for the rescue effects and for Rab26 activation (eg. Empty vector and wild-type or inactive Rab26). One of the main conclusions is that "This secretory pathway depends on the activation of the small GTPase Rab26 by *Plekhg5*", so this statement requires better support.

We followed the reviewer's comment and included an additional control. We knocked down Plekhg5 and simultaneously expressed wild-type Rab26. In contrast to the expression of constitutive active Rab26, wild-type Rab26 did not rescue the reduced levels of Sod1 in the medium of Plekhg5 depleted cells. These data further support our hypothesis that this pathway depends on the activation of Rab26 by Plekhg5. In the revised version of the manuscript, the new data are included in Figure 2 L-N.

Figure 2

(L) Western blot of primary MN lysates showing the lentiviral knockdown of Plekhg5 and simultaneous expression of Flag-Rab26-WT or Flag-Rab26-QL.

(M) Expression of Flag-Rab26-QL restores the reduced Sod1 medium levels in Plekhg5-deficient cells as shown by Western blot.

(N) Western blot quantifications of the Sod1 intensities upon knockdown of Plekhg5 and simultaneous expression of Flag-Rab26-WT or Flag-Rab26-QL in the lysate and media of primary MNs. sh-Luc, n=12; sh-Plekhg5-E, n=12; sh-Plekhg5-E+Flag-Rab26-WT, n=5, sh-Plekhg5-E+Flag-Rab26-QL, n=12. One-way ANOVA; Holm-Šidák's multiple comparisons test.

5. Figure 3 nicely shows dependence on Atg5, and somewhat on Atg9, for secretion of Sod1 into the medium. To determine whether this mechanism also includes distal components of secretory autophagy, it would be useful to similarly test STX3, STX4 or SNAP29.

We followed the reviewers suggestion and tested the involvement of distal secretory autophagy components in this pathway. Due to the overlap with the suggestion of reviewer 1, we genetically targeted SNAP29. We also targeted STX17 (fusion with lysosomes) and SNAP23 (fusion with the plasma membrane). sh-RNA-mediated knockdown of each of these SNARE proteins in primary motoneurons resulted in reduced levels of Sod1 in the cell culture medium, indicating that distal components of secretory autophagy are included in this pathway, and that the secretion of Sod1 requires the fusion between autophagosome and lysosome. We incorporated this new data set into the revised version of Fig. 4 (see below).

Figure 4

J-L: Western blots showing the knockdown of Stx17 (J), Snap29 (K), and Snap23 (L) upon simultaneous lentiviral expression of two different sh-RNAs in primary MNs.

M-O: The Sod1 secretion is blocked upon knockdown of Stx17 (M), Snap29 (N), and Snap23 (O). Western blots of lysates and media of primary MNs transduced with sh-RNAs targeting the indicated SNAREs.

P: Quantification of the Sod1 intensity in lysates and media. sh-Stx17, n=4; sh-Snap29, n=6; sh-Snap23, n=6. One-sample t-test.

All data are shown as mean \pm SEM. *p % 0.05; ***p % 0.001.

6. P. 7, last paragraph would benefit from clarification. "In this scenario, Sod1 would be released as a free protein". Since the previous sentence is referring to fusion of the outer autophagosomal membrane with a lysosome, does this mean "released into the autolysosome"? Rather, I think the authors intend to indicate released into the media as a free protein versus released into the media within a sEV.

We thank the reviewer for pointing out this critical point. We revised this sentence to clarify this paragraph. Now, it reads as following: "In this scenario, Sod1 would be released into the media as a free protein, and not within an extracellular vesicle."

7. Figure 5B: BafA1 experiment: if BafA1 blocks the fusion of autophagosomes with lysosomes (or LROs), then what is the proposed mechanism (ie. secretory route) for the enhanced Sod1 secretion upon treatment with BafA1? This finding appears inconsistent with the authors proposed model.

BafA1 primarily blocks the acidification of the lysosome by inhibiting the ATPase. As a secondary, later effect, fusion of autophagosomes and lysosomes is inhibited [9]. Since we only measured the accumulation of Sod1 in the medium at the endpoint, it is difficult to determine whether the effect occurs early, late, or continuously. To overcome this limitation, we used Pepstatin A and E64D as specific inhibitors to block lysosomal function and not the fusion with autophagosomes. With this approach, we obtained similar results showing an enhanced secretion of Sod1 (Fig. 5 B, C). These results suggest that lysosomal dysfunction triggers the secretion of Sod1 and not the inhibition of the fusion between autophagosomes and lysosomes.

Furthermore, genetic targeting of Stx17, which is essential for autophagosome and lysosome fusion, blocks the secretion of Sod1, confirming that fusion between both organelles precedes the fusion with

the plasma membrane. In conclusion, these data suggest that the effect of an enhanced Sod1 secretion upon BafA1 exposure occurs early due to an impaired lysosomal acidification.

In the revised version of the manuscript, we explained this section in more detail as follows: “Therefore, we analyzed the secretion of Sod1 in NSC34 cells upon lysosomal disruption by the V-ATPase inhibitor Bafilomycin A1, which blocks the acidification of lysosomes (Fig. 5 A). As a secondary, late effect, Bafilomycin exposure also blocks the fusion of autophagosomes and lysosomes [9]. To more specifically inhibit lysosomal function, and not fusion between autophagosomes and lysosomes, we applied a combination of the aspartic protease inhibitor Pepstatin A and the cysteine protease inhibitor E64D (Fig. 5 B). Indeed, we detected an enhanced Sod1 secretion upon blockage of lysosomal acidification or inhibition of lysosomal proteolysis, which indicates that Sod1 is secreted in response to lysosomal dysfunction (Fig. 5 C).”

8. The rationale and conclusions regarding Figure 5E-F are unclear. The authors conclude that these images show that p62 and ubiquitin were absent from Lamp1+ vesicle clusters, “confirming that the global proteostasis is unaffected by the depletion of Plekhg5”. From the images, it appears that p62 and Lamp1+ co-localize in some instances. The description and/or images require clarification and also quantitation.

We followed the reviewer’s advice and quantified the co-localization. Furthermore, we added a new paragraph for a better description of the rationale and conclusions. The quantifications confirm that p62 and Ubiquitin are absent from the Lamp1+ vesicle accumulation. In the revised version of figure 5, we show more representative images.

The results are now described as follows: “To confirm that the Lamp1⁺ vesicle clusters do not represent an accumulation of dysfunctional lysosomes, we stained for the autophagy receptor p62 and Ubiquitin. Both proteins accumulate upon lysosomal dysfunction as previously described [10, 11]. The absence of both, p62 and Ubiquitin from Lamp1⁺ clusters confirms that depletion of Plekhg5 did not result in lysosomal dysfunction.”

9. Figure 6L requires quantitation, especially to substantiate the difference between Plekhg5^{+/+}SOD1[G93A] and Plekhg5^{-/-}SOD1[G93A].

Quantifications are included in the revised version of the manuscript and confirm a reduced microglia activation in Plekhg5^{-/-}SOD1[G93A] compared to Plekhg5^{+/+}SOD1[G93A] mice. The data are shown in Fig. 6 M.

Figure 6

(M) Quantification of the CD68 immunoreactivity area. Plekhg5^{+/+}, n=3; Plekhg5^{-/-}, n=5; Plekhg5^{+/+} Sod1^{G93A}, n=4; Plekhg5^{-/-} Sod1^{G93A}, n=3. One-way ANOVA; Tukey’s Multiple Comparisons.

Other:

p.31. Figure 2 legend is missing a title.

We introduced a title.

Figure 6D: y axis label should be survival, not onset.

Thanks for pointing out this mistake. It’s corrected in the revised version of the manuscript.

References:

- Schweingruber, C. and E. Hedlund, *The Cell Autonomous and Non-Cell Autonomous Aspects of Neuronal Vulnerability and Resilience in Amyotrophic Lateral Sclerosis*. Biology (Basel), 2022. **11**(8).
- Xie, Y.X., et al., *Lysosomal exocytosis releases pathogenic alpha-synuclein species from neurons in synucleinopathy models*. Nat Commun, 2022. **13**(1): p. 4918.
- Trajkovic, K., H. Jeong, and D. Krainc, *Mutant Huntingtin Is Secreted via a Late Endosomal/Lysosomal Unconventional Secretory Pathway*. J Neurosci, 2017. **37**(37): p. 9000-9012.
- Kang, S., et al., *Autophagy-Mediated Secretory Pathway is Responsible for Both Normal and Pathological Tau in Neurons*. J Alzheimers Dis, 2019. **70**(3): p. 667-680.
- Hung, S.T., et al., *PIKFYVE inhibition mitigates disease in models of diverse forms of ALS*. Cell, 2023. **186**(4): p. 786-802 e28.
- Cruz-Garcia, D., et al., *A diacidic motif determines unconventional secretion of wild-type and ALS-linked mutant SOD1*. J Cell Biol, 2017. **216**(9): p. 2691-2700.
- Zhang, M., et al., *Translocation of interleukin-1beta into a vesicle intermediate in autophagy-mediated secretion*. Elife, 2015. **4**.

8. Villeneuve, J., et al., *Unconventional secretion of FABP4 by endosomes and secretory lysosomes*. J Cell Biol, 2018. **217**(2): p. 649-665.
9. Klionsky, D.J., et al., *Does bafilomycin A1 block the fusion of autophagosomes with lysosomes?* Autophagy, 2008. **4**(7): p. 849-50.
10. Settembre, C., et al., *A block of autophagy in lysosomal storage disorders*. Hum Mol Genet, 2008. **17**(1): p. 119-29.
11. Thelen, M., et al., *Disruption of the autophagy-lysosome pathway is involved in neuropathology of the nclf mouse model of neuronal ceroid lipofuscinosis*. PLoS One, 2012. **7**(4): p. e35493.

REVIEWERS' COMMENTS

Reviewer #1 (Remarks to the Author):

The authors have satisfactorily answered the reviewers' comments.

Reviewer #2 (Remarks to the Author):

The authors have done a splendid job to address the questions I asked. no further issues.

Reviewer #3 (Remarks to the Author):

The authors addressed comments thoroughly, and the addition of new data and quantitation have strengthened the manuscript.

I have just one minor point for clarification:

p. 13 new highlighted text:

“As recently shown, this contrasts TDP43 and Tau, which are detectable in plasma extracellular vesicles, providing a promising invasive biomarker for frontotemporal dementia and ALS [66]. “

What is meant by “invasive biomarker”? Is this an error?

Response to the REVIEWERS' COMMENTS

We thank all three reviewers for their positive feedback.

Reviewer #1 (Remarks to the Author):

The authors have satisfactorily answered the reviewers' comments.

Reviewer #2 (Remarks to the Author):

The authors have done a splendid job to address the questions I asked. no further issues.

Reviewer #3 (Remarks to the Author):

The authors addressed comments thoroughly, and the addition of new data and quantitation have strengthened the manuscript.

I have just one minor point for clarification:

p. 13 new highlighted text:

“As recently shown, this contrasts TDP43 and Tau, which are detectable in plasma extracellular vesicles, providing a promising invasive biomarker for frontotemporal dementia and ALS [66]. “

What is meant by “invasive biomarker”? Is this an error?

We thank the reviewer for pointing out this mistake. It was supposed to mean “non-invasive”. We revised the manuscript accordingly.